*Report*

# The actin binding protein profilin 1 localizes inside mitochondria and is critical for their function

Tracy-Ann Read [1,11]✉, Bruno A Cisterna [1,11], Kristen Skruber [2], Samah Ahmadieh [3,4], Tatiana M Liu [1], Josefine A Vitriol[1], Yang Shi [1,5], Joseph B Black[6], Mitchell T Butler [7,8], Halli L Lindamood [1], Austin EYT Lefebvre [9], Alena Cherezova [10], Daria V Ilatovskaya [10], James E Bear[7,8], Neal L Weintraub [3,4] & Eric A Vitriol [1]✉

## Abstract

**The monomer-binding protein profilin 1 (PFN1) plays a crucial role in actin polymerization. However, mutations in PFN1 are also linked to hereditary amyotrophic lateral sclerosis, resulting in a broad range of cellular pathologies which cannot be explained by its primary function as a cytosolic actin assembly factor. This implies that there are important, undiscovered roles for PFN1 in cellular physiology. Here we screened knockout cells for novel phenotypes associated with PFN1 loss of function and discovered that mitophagy was significantly upregulated. Indeed, despite successful autophagosome formation, fusion with the lysosome, and activation of additional mitochondrial quality control pathways, PFN1 knockout cells accumulate depolarized, dysmorphic mitochondria with altered metabolic properties. Surprisingly, we also discovered that PFN1 is present inside mitochondria and provide evidence that mitochondrial defects associated with PFN1 loss are not caused by reduced actin polymerization in the cytosol. These findings suggest a previously unrecognized role for PFN1 in maintaining mitochondrial integrity and highlight new pathogenic mechanisms that can result from PFN1 dysregulation.**

**Keywords** Profilin; Actin; Mitochondria; Mitophagy; Mitochondrial-derived Vesicles
**Subject Categories** Cell Adhesion, Polarity & Cytoskeleton; Membranes & Trafficking

## Introduction

The actin cytoskeleton is a dynamic structural network within eukaryotic cells which regulates their shape and behavior. It is indispensable for driving cellular processes including motility, cytokinesis, and intracellular communication (Blanchoin et al, 2014; Pollard and Borisy, 2003; Skruber et al, 2018). Inside the cell, forces generated by the assembly and organization of actin filaments are used to control the dynamics of organelles like mitochondria (Fung et al, 2019; Manor et al, 2015). Proper mitochondrial function requires precise control over the shape and distribution of mitochondria networks, which is achieved through mitochondria fission, fusion, transport, and clearance of damaged organelles (Gatti et al, 2023; Korobova et al, 2013; Quintero et al, 2009). Actin has been implicated in all of these regulatory steps, largely through the polymerization of filaments on the mitochondria surface or contact sites on the endoplasmic reticulum (Illescas et al, 2021; Tilokani et al, 2018). For example, INF2/Spire1C and Arp2/3-mediated filament assembly are essential components of the fission process (Fung et al, 2019; Manor et al, 2015). Moreover, tethering of mitochondria to actin filaments via Miro or Myosin 19 can alter mitochondria transport and physiology through morphological changes (Lopez-Domenech et al, 2018; Quintero et al, 2009).

Actin filaments are also crucial for metabolic pathways that in turn regulate mitochondrial function. For instance, activation of the glycolytic enzymes' aldolase and glyceraldehyde phosphate dehydrogenase, can occur by direct binding to F-actin (Ouporov et al, 2001; Ouporov et al, 1999). Furthermore, TRIM21 (Tripartite Motif-containing Protein 21) is sequestered by F-actin bundles, thereby reducing its access to substrates such as the rate-limiting metabolic enzyme phosphofructokinase (PFK), thus ensuring high glycolytic rates (Park et al, 2020). In neurons, actin in mitochondria is needed for the retention of cytochrome c between respiratory chain complexes III and IV through direct association with both complexes (Takahashi et al, 2018). In addition, actin

[1]Department of Neuroscience and Regenerative Medicine, Medical College of Georgia at Augusta University, Augusta, GA, USA. [2]Department of Cellular and Molecular Pharmacology, University of California San Francisco, San Francisco, CA, USA. [3]Vascular Biology Center, Medical College of Georgia, Augusta University, Augusta, GA, USA. [4]Department of Medicine, Medical College of Georgia at Augusta University, Augusta, GA, USA. [5]Department of Population Health Sciences, Medical College of Georgia at Augusta University, Augusta, GA, USA. [6]Division of Urologic Surgery, Beth Israel Deaconess Medical Center, Boston, MA, USA. [7]Department of Cell Biology and Physiology, University of North Carolina at Chapel Hill School of Medicine, Chapel Hill, NC, USA. [8]Lineberger Comprehensive Cancer Center, University of North Carolina at Chapel Hill School of Medicine, Chapel Hill, NC, USA. [9]Calico Life Sciences, South San Francisco, CA, USA. [10]Department of Physiology, Medical College of Georgia at Augusta University, Augusta, GA, USA. [11]These authors contributed equally: Tracy-Ann Read, Bruno A Cisterna. ✉E-mail: tread@augusta.edu; evitriol@augusta.edu

 

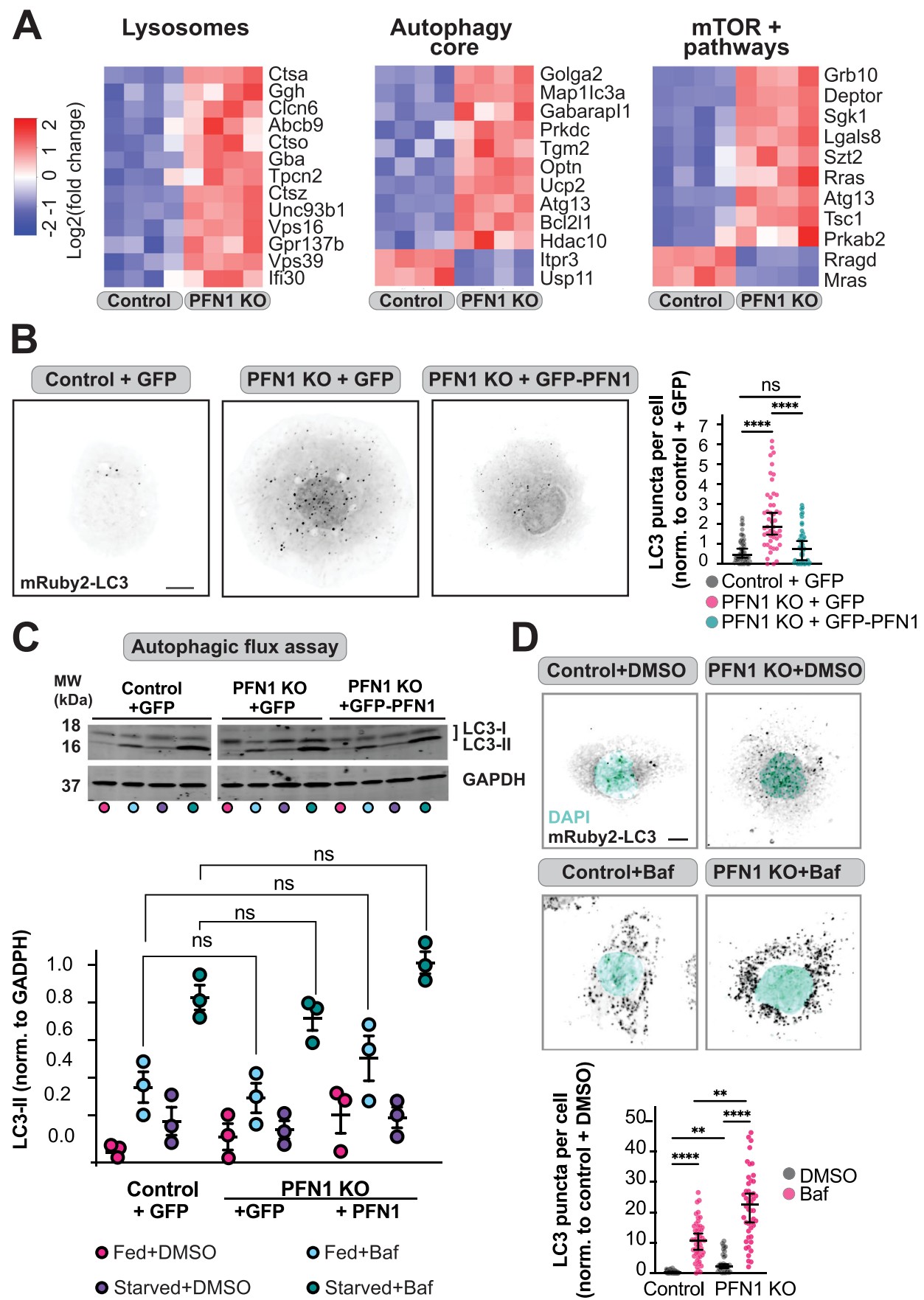

**Figure 1.  Loss of PFN1 causes an upregulation of autophagy.**

(A) Heat maps of RNA-seq analysis performed on control and PFN1 KO cells, showing differential expression of genes involved in autophagy, endosome/lysosome and mTOR signaling pathways. Log2 fold change scores are colored according to the key on left. (B) Representative images of mRuby2-LC3 in control and PFN1 KO cells expressing either GFP or GFP-PFN1 (left) and quantification of LC3 puncta (right). Each data point on the graph represents the number of LC3 puncta counted in one cell ($n = 56$ cells for Control + GFP, $n = 51$ cells for PFN1 KO + GFP, and $n = 54$ cells for PFN1 KO + GFP-PFN1). Data is shown as median ± 95% CI. Significance was calculated against Control + GFP using ANOVA and Dunnett's post hoc test. Scale bar = 10 μm. (C) Western blot of LC3 and GAPDH in control and PFN1 KO cells expressing either GFP or GFP-PFN1 (top). Cells were given normal media (fed) or were nutrient-deprived (starved) for 6 h and were treated with Bafilomycin A1 (Baf) to inhibit lysosome-mediated degradation or DMSO vehicle control for 4 h. Quantification of autophagic flux where LC3II levels were normalized against GAPDH (bottom). Data is shown as mean ± SEM and each data point is one biological replicate ($n = 3$). Significance was calculated using ANOVA and Tukey's post hoc test. (D) Representative images of mRuby2-LC3 and DAPI in control and PFN1 KO cells treated with Bafilomycin A1 (Baf) or DMSO vehicle control for 4 h (top) and quantification of LC3 puncta (bottom). Each data point on the graph represents the number of LC3 puncta counted in one cell ($n = 64$ cells for Control + DMSO, $n = 50$ cells for Control + Baf, $n = 39$ cells for PFN1 KO + DMSO, and $n = 48$ cells for PFN1 KO + Baf). Data is shown as median ± 95% CI. Significance was calculated using a Kruskal–Wallis test followed by Dunn's multiple comparisons test. Scale bar = 10 μm. Data information: ****$p < 0.0001$, **$p < 0.01$, ns $p > 0.05$. Source data are available online for this figure.

depolymerization with cytochalasin b can enhance mitochondrial respiration through increased complex IV activity (Takahashi et al, 2018).

Here, we investigate the role of profilin 1 (PFN1) in controlling mitochondria function. PFN1 is an actin monomer-binding protein with a strong influence over actin filament assembly. Profilin prevents spontaneous filament assembly, promotes the exchange of ADP-actin to the polymerization competent ATP-bound form, and can determine which filaments assemble from the monomer pool through interactions with different actin polymerases (Rotty et al, 2015; Skruber et al, 2020). Apart from mitochondria dysfunction associated with amyotrophic lateral sclerosis (ALS) linked mutants of PFN1 (Fil et al, 2017; Teyssou et al, 2022; Yang et al, 2016), no current evidence indicates a direct connection between PFN1 and mitochondria. However, when we deleted the *PFN1* gene in two different cell lines, we found that PFN1 is critical for mitochondrial integrity, with the loss of PFN1 causing a chronic activation of quality control pathways due to the accumulation of depolarized, abnormally shaped mitochondria. Most unexpectedly, we provide compelling evidence that PFN1 is found inside mitochondria. ALS-associated PFN1 mutants can also aggregate inside mitochondria, potentially causing additional damage. Thus, not only is PFN1 a critical regulator of mitochondria, but dysregulation of PFN1 can have drastic effects on mitochondria function.

## Results and discussion

### Loss of PFN1 causes an upregulation of autophagy

In previous work we used CRISPR/Cas9 to knock out (KO) PFN1 in the central nervous system-derived Cath.a-differentiated (CAD) cell line (Skruber et al, 2020). Interestingly, RNA-seq analysis (Data ref: Skruber et al, 2020) identified significant changes in expression of genes associated with lysosome/endosome systems and autophagy (Bordi et al, 2021) upon the loss of PFN1 expression (Fig. 1A). This included upregulation of genes in the lysosomal proteolytic system (*CTSZ, CTSA*), genes in the endosome/lysosome pathway (*TPCN2, UNC93B1, VPS16, VPS9*), and autophagy activating genes such as *ATG13, USP11, HDAC10,* and *PRKDC*. Furthermore, PFN1 KO cells had differential expression of mammalian target of rapamycin (mTOR) pathway deactivating genes (*GRB10, DEPTOR, TSC1, RRAGD*) (Fig. 1A), which is the predominant signaling event

that induces autophagy (Bordi et al, 2021). We confirmed the upregulation of *DEPTOR* in both fed and starved conditions compared to control cells (Fig. EV1A) and a decrease in phosphorylated mTOR (Fig. EV1B) in PFN1 KO cells via quantitative western blotting. We also expressed mRuby2-LC3, which forms puncta when autophagosomes are formed (Fujita et al, 2008; Kabeya et al, 2000), and found a two-fold increase in LC3 puncta in PFN1 KO cells (Fig. 1B). Importantly, this phenotype could be rescued by expressing a GFP-PFN1 construct (Fig. 1B). Thus, autophagy is significantly upregulated in the absence of PFN1.

The increase in autophagosomes observed in PFN1 KO cells could either be explained by a defect in protein and organelle homeostasis, resulting in decreased turnover, or an increased need for autophagy. To test if autophagy was still functional without PFN1, we performed autophagic flux assays. Quantitative western blot analysis measuring the amounts of soluble LC3-I and lipid bound LC3-II was used to determine autophagosome formation after macro-autophagy was induced by nutrient deprivation. In addition, autophagosome production was measured by adding Bafilomycin A1 to inhibit autophagosome fusion with the lysosome and prevent their turnover (Yoshii and Mizushima, 2017). As shown in Fig. 1C, autophagic flux was comparable in control, PFN1 KO, and PFN1 KO cells rescued with GFP-PFN1. We also used Bafilomycin A1 in the mRuby2-LC3 microscopy assays to measure autophagosome turnover in fed conditions (Fig. 1D) and found that the relative increase in autophagosomes after addition of Bafilomycin A1 was the same in control and PFN1 KO cells, again indicating that there was no general deficiency in autophagosome biogenesis or turnover following PFN1 depletion.

### Autophagy caused by the loss of PFN1 selectively targets mitochondria

Since macro-autophagy appeared to be unaffected by PFN1 depletion (Fig. 1C), we next investigated whether the accumulation of autophagic vesicles in PFN1 KO cells was caused by an upregulation of selective autophagy. A clue came from the RNA-seq data, which revealed that several major modulators of mitophagy were differentially expressed upon the loss of PFN1 (Data ref: Skruber et al, 2020). This included increased expression of *TGM2, OPTN,* and *BCL2L1*, and downregulation of the mitophagy inhibitor *SIAH3* (Gladkova et al, 2018; Lazarou et al, 2015;

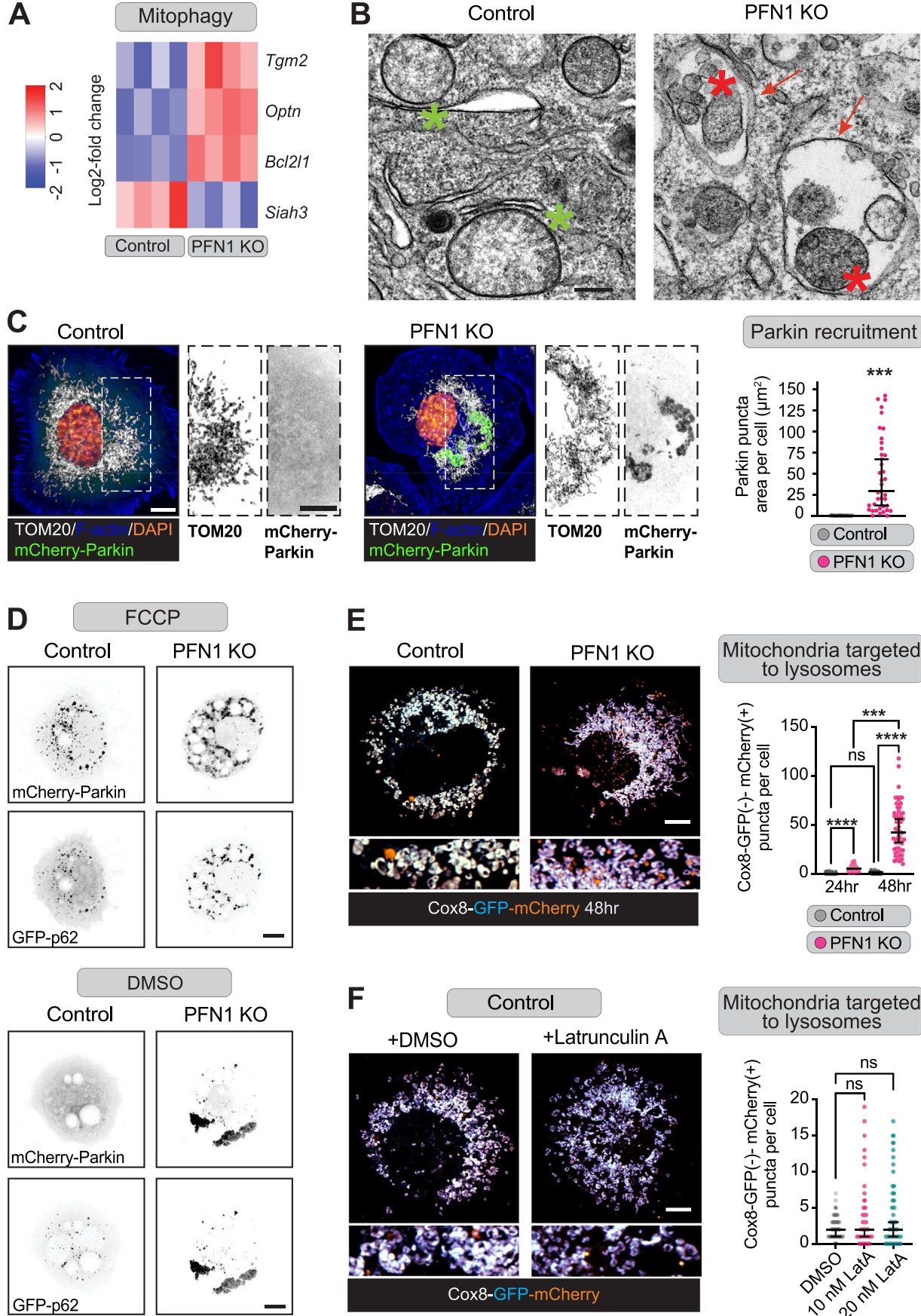

Figure 2.  Autophagy induced by the loss of PFN1 selectively targets mitochondria.

(A) Heat map of RNA-seq analysis performed on control and PFN1 KO cells, showing differential expression of genes involved in mitophagy. Log2 fold change scores are colored according to the key on left. (B) Representative transmission electron micrographs of control and PFN1 KO cells. Control cells show normal mitochondria (green stars) while PFN1 KO cells have multiple double membrane bound autophagic vesicles (red arrows) containing mitochondria (red stars), scale bar = 1 μm. (C) Representative images of control and PFN1 KO cells expressing mCherry-Parkin that were also stained for TOM20, F-actin, and DAPI (left). White dashed boxes indicate the region used for the enlarged single channels images. Quantification of Parkin puncta (right). Each data point represents the total area of Parkin puncta measured in one cell (n = 40 for both Control and PFN1 KO cells). Data is shown as median ± 95% CI. Significance was calculated using a Mann–Whitney test. Scale bar = 10 μm. (D) Representative images of Control and PFN1 KO cells expressing mCherry-Parkin and GFP-p62 treated with 20 μM of the mitochondrial uncoupling agent FCCP (top) or an equivalent amount of DMSO vehicle control (bottom) for 20 min. Scale bar = 10 μm. (E) Representative images of Control and PFN1 KO cells expressing the mitophagy reporter Cox8-GFP-mCherry 48 h post electroporation (left). When Cox8-GFP-mCherry enters an acidified environment, the GFP fluorescence is quenched whilst the mCherry fluorescence remains intact. The lookup tables used for GFP and mCherry are white when overlapped. Quantification of Cox8 puncta at 24 and 48 h post electroporation that only had mCherry fluorescence (right). Each data point on the graph represents the number of Cox8-GFP(-)-mCherry(+) puncta that were counted in one cell (n = 60 for all conditions). Data are shown as median ± 95% CI. Significance was calculated using a Kruskal–Wallis test followed by Dunn's multiple comparisons test. Scale bar = 10 μm. (F) Representative images of Control cells expressing Cox8-mCherry-GFP treated with 20 nM Latrunculin A (Lat A) overnight or DMSO vehicle control (left). This low dose Lat A reduces F-actin to 30–40% without altering cell morphology or causing cell death. Lookup tables used for GFP and mCherry are white when overlapped. Quantification of Cox8 puncta that only had mCherry fluorescence (right). Each data point on the graph represents the number of Cox8-GFP(-)-mCherry(+) puncta were counted in one cell (n = 80 for DMSO, n = 81 for 10 nM Lat A, and n = 86 for 20 nM Lat A). Data shown as median ± 95% CI. Significance was calculated using a Kruskal–Wallis test followed by Dunn's multiple comparisons test. Scale bar = 10 μm. Data information: ****$p < 0.0001$, ns $p > 0.05$. Source data are available online for this figure.

Narendra et al, 2008; Pickrell and Youle, 2015) (Fig. 2A). Transmission electron microscopy imaging confirmed the activation of mitophagy in PFN1 KO cells, which contained abundant double membrane bound autophagic vesicles containing mitochondria (Fig. 2B).

Presently, the best described mitophagy pathway involves PTEN-induced kinase 1 (PINK1), which activates the E3 ubiquitin ligase Parkin to tag depolarized and damaged mitochondria for degradation (Gladkova et al, 2018; Lazarou et al, 2015; Narendra et al, 2008; Pickrell and Youle, 2015). Siah3 inhibits PINK1/Parkin translocation to damaged mitochondria (Abd Elghani et al, 2022; Hasson et al, 2013), and its downregulation in PFN1 KO cells (Fig. 2A) suggested an increase in PINK1/Parkin-based mitophagy. Indeed, we found a 25-fold increase in the amount of mCherry-Parkin that accumulated on mitochondria in PFN1 KO compared to Control cells (Fig. 2C), in a similar pattern to that observed when cells were treated with 10 μM FCCP to depolarize mitochondria (Fig. 2D). In addition, Parkin and the autophagy adapter p62/SQSTM1 colocalized at mitochondria in both FCCP treated cells and untreated PFN1 KO cells (Fig. 2D), indicating that Parkin labeled mitochondria in both conditions are being targeted for autophagic degradation.

To determine if mitochondria in PFN1 KO cells are successfully targeted for lysosomal degradation after Parkin recruitment, we used the inner mitochondrial membrane (IMM) targeted mitophagy reporter Cox8-GFP-mCherry (Ma and Rojansky et al, 2016). The GFP fluorescence of the probe is rapidly quenched in acidified compartments while the mCherry fluorescence remains intact, which reports mitochondria that are delivered to autolysosomes (Ma and Ding, 2021; Rojansky et al, 2016). Using this approach, we found a significant increase in mCherry-only puncta in PFN1 KO cells compared to controls, which further increased to a 10-fold difference after 48 h of probe expression (Fig. 2E). Therefore, autophagy is able to initiate (Figs. 1D–F and 2C) and mitochondria are successfully targeted to lysosomes for degradation in PFN1's absence. Interestingly, no change was observed in the number of mitochondria in acidified compartments in control cells between 24 and 48 h, indicating a continuous clearance of these structures. The

accumulation of lysosome-delivered mitochondria in PFN1 KO cells over time suggests that they may harbor clearance defects beyond the stage of lysosomal activation.

Given that the actin cytoskeleton is important for mitochondrial homeostasis, we next sought to determine if mitophagy was activated in response to the large decrease in polymerized actin previously observed in PFN1 KO cells (Skruber et al, 2020). However, reducing the amount of polymerized actin in control cells by 30–40% with a low overnight dose of Latrunculin A (10–20 nM) to approximate the loss of F-actin caused by PFN1 KO (Cisterna et al, 2024) did not result in increased mitochondria delivery to lysosomes (Fig. 2F). Therefore, prolonged actin depolymerization by itself is not sufficient to damage mitochondria and activate mitophagy. This was surprising, since dysregulation of autophagy in response to actin depolymerization has been reported by others (Aguilera et al, 2012; Aplin et al, 1992; Lee et al, 2010; Mi et al, 2015; Reggiori et al, 2005; Zhuo et al, 2013). Though, an important difference in our study is that we measured autophagy/mitophagy under conditions where actin was partially depolymerized overnight to replicate a physiological defect in actin assembly, whereas previous work used short term, high concentrations of Latrunculin A to depolymerize the majority of actin filaments.

In addition, there has been compelling evidence linking actin polymerization on the mitochondria surface to mitophagy. For example, cofilin potentiates PINK1/PARK2-dependent mitophagy (Li et al, 2018) and MYO6 (myosin VI) forms a complex with PARK2 and initiates the assembly of F-actin cages to encapsulate damaged mitochondria (Kruppa and Buss, 2018; Kruppa et al, 2018). Mitophagy also requires fission of the damaged portion of the organelle, and several fission pathways are known to require polymerization of actin on the mitochondrial surface (Coscia et al, 2023; Gatti et al, 2023), including those dependent on the formin INF2, which requires profilin-actin to assemble filaments (Fung et al, 2019; Gurel et al, 2015; Manor et al, 2015). Here, either these pathways are not affected by a substantial loss in the ability to polymerize actin, or PFN1 may have a more complex role in maintaining mitochondrial integrity than simply polymerizing actin on the outer mitochondrial membrane (OMM).

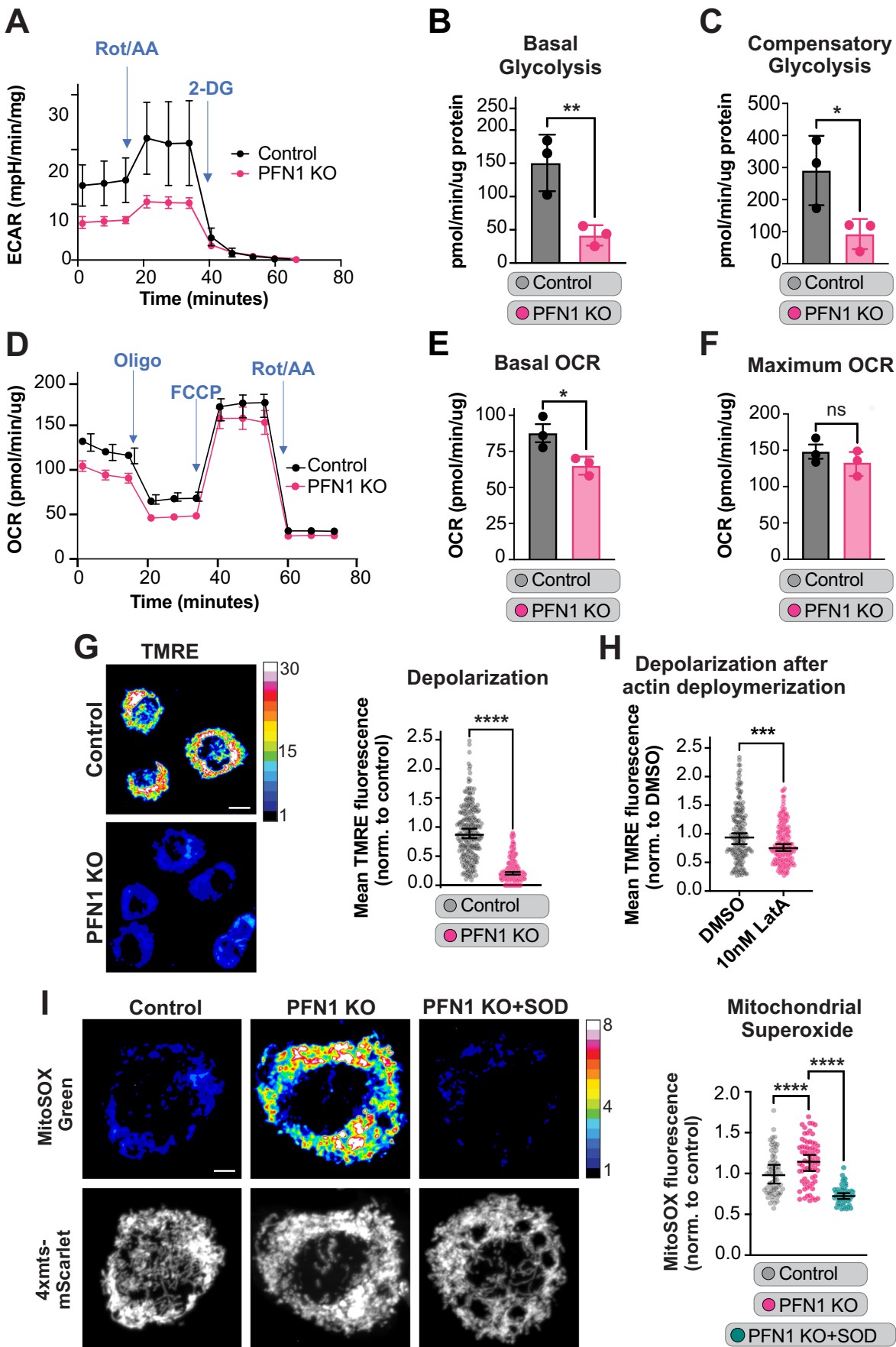

◀   **Figure 3.  Loss of PFN1 disrupts mitochondrial metabolism.**

(A–F) Seahorse Extracellular Flux Analyzer measurements of glycolysis (A–C) and oxidative phosphorylation (D–F). Glycolysis is measured using the extracellular acidification rate (ECAR) with the Seahorse XF Glycolysis Stress Test and oxidative phosphorylation is measured using the oxygen consumption rate (OCR) with the Seahorse XF Mito Stress Test. (A) Time course of Control and PFN1 KO cells in the Seahorse XF Glycolysis Stress Test. The data is displayed as mean ± SEM and each data point is one biological replicate ($n = 3$). (B, C) Quantification of basal (B) and compensatory glycolysis (C) from (A). The data is displayed as mean ± SD and each data point is one biological replicate ($n = 3$). Significance was calculated using a Student's t-test. (D) Time course of Control and PFN1 KO cells in the Seahorse XF Mito Stress Test. The data is displayed as mean ± SEM and each data point is one biological replicate ($n = 3$). (E, F) Quantification of basal (E) and maximum OCR (F) from (D). The data is displayed as mean ± SD and each data point is one biological replicate ($n = 3$). Significance was calculated using a Student's t-test. (G) Representative images of Control and PFN1 KO cells stained with TMRE, a fluorogenic dye for measuring membrane potential in live cells (left). The images are pseudo colored using the lookup table on the right and scaled identically so they can be compared. Quantification of mean TMRE fluorescence (right). Each data point represents the mean TMRE fluorescence of one cell ($n = 250$ cells for Control and $n = 231$ cells for PFN1 KO). Significance was calculated using a Mann–Whitney test. Scale bar = 10 μm. (H) Quantification of mean TMRE fluorescence in Control cells treated overnight with 10 nM Latrunculin A (LatA) or DMSO vehicle control. Each data point represents the mean TMRE fluorescence of one cell ($n = 178$ cells for DMSO and $n = 197$ cells for 10 nM Lat A). Significance was calculated using a Mann–Whitney test. (I) Representative images of Control and PFN1 KO cells expressing 4xmts-mScarlet to label mitochondria and stained with the MitoSOX Green superoxide probe (left). Superoxide dismutase (SOD) was also added to PFN1 KO cells to remove superoxide at the mitochondrial membrane. The images are pseudo colored using the lookup table on the right and scaled identically so they can be compared. Quantification of mean MitoSOX Green fluorescence (right). Each data point represents the mean TMRE fluorescence of one cell ($n = 66$ cells for Control, $n = 70$ for PFN1 KO, and $n = 67$ cells for PFN1 KO + SOD). Significance was calculated using ANOVA and Tukey's post hoc test. Scale bar = 5 μm. Data information: ****$p < 0.0001$, ***$p < 0.001$, **$p < 0.01$, *$p < 0.05$, ns $p > 0.05$. Source data are available online for this figure.

## Loss of PFN1 disrupts mitochondrial metabolism, morphology, and dynamics

Having determined that loss of PFN1 induces mitophagy, we next examined the cause of targeted mitochondria degradation. First, we assessed metabolic function using the Seahorse Extracellular Flux Analyzer which measures glycolysis and oxidative phosphorylation indirectly through the extracellular acidification rate (ECAR) oxygen consumption rate (OCR), respectively. These assays revealed that PFN1 KO cells have deficient basal and compensatory glycolysis compared to Control cells, as well as a slight reduction in basal respiration (Fig. 3A–F).

Given that the loss of mitochondrial membrane potential often precedes mitophagy (Narendra et al, 2008), we next assessed if mitochondrial membrane potential was altered in PFN1 KO cells using TMRE (tetramethylrhodamine, ethyl ester), a cell permeant dye that accumulates in active mitochondria and fails to sequester in depolarized or inactive mitochondria. PFN1 KO cells had significantly lower fluorescence of TMRE-labeled mitochondria compared to Control cells (Fig. 3G). Further investigation also showed that mitochondrial depolarization in PFN1 KO cells is not a byproduct of general actin depolymerization, as treating Control cells overnight with Latrunculin A only caused a mild mitochondrial depolarization which did not approximate the huge loss of TMRE labeling found in PFN1 KO cells (Fig. 3H). Finally, we used MitoSox Green to measure superoxide release at the mitochondrial membrane, a common byproduct of depolarization (Mailloux, 2020), and found that PFN1 KO cells had elevated mitochondrial superoxide (Fig. 3I). Along with the Seahorse data, these experiments demonstrate that PFN1 depletion impairs mitochondrial function and help explain why mitochondria quality control pathways are activated in PFN1 KO cells.

We next investigated whether loss of PFN1 expression affected mitochondria morphology. Transmission electron microscopy images showed distinct differences in mitochondria structure in PFN1 KO cells, namely elongated mitochondrial networks with dysmorphic cristae and many accounts of formation and release of mitochondrial-derived vesicles (MDVs) (Sugiura et al, 2014) (Fig. 4A,D). To quantify these morphological features, we performed immunocytochemistry of the OMM protein TOM20

and imaged the cells using optical pixel reassignment (SoRa) super-resolution spinning disk confocal microscopy. 3D segmentation of the confocal z-stacks allowed us to obtain the size, shape, and position of all mitochondria present in the cell (Fig. 4B). As shown in Fig. 4A–C, PFN1 KO mitochondria were more elongated and present in larger networks than Control cells. PFN1 KO cells also had a higher sum volume of mitochondria (Fig. 4C), which as previously mentioned, is likely caused by a clearance defect of lysosome-targeted mitochondria (Fig. 2E). Similar aberrant mitochondria morphology was also observed in live cells expressing 4xmts-mNeonGreen to label the intermembrane space (Chertkova et al, 2020) (Fig. EV2A,B). Re-expressing wild-type PFN1 rescued the total volume phenotype whereas the R88E-PFN1 mutant deficient in binding actin did not (Fig. EV2C), indicating that PFN1's regulation of actin is important for mitochondrial clearance. Interestingly, ALS-linked PFN1 mutants expressed in PFN1 KO cells are unable to restore mitochondrial phenotypes such as elongation and total volume, even in cells that lacked visible protein aggregates (Fig. EV2D). This was also the case for the E117G mutant, which is considered an ALS risk factor with only mild actin assembly defects (Cisterna et al, 2024). Overexpressing mCherry-Parkin for 48 h to force additional mitochondrial degradation, did not decrease total mitochondrial volume in PFN1 KO cells (Fig. EV2E), further supporting a possible late-stage defect in the removal of damaged mitochondria.

MDVs are small vesicles which direct damaged mitochondrial components to the cell's degradative machinery that can be upregulated when autophagy is compromised to maintain mitochondrial homeostasis (Sugiura et al, 2014; Towers, 2021; Towers et al, 2021). To confirm the increase in MDV production in PFN1 KO cells suggested by electron microscopy (Fig. 4A–D), we measured all TOM20-positive mitochondria fragments that met the MDV size criteria (Sugiura et al, 2014; Towers et al, 2021) and found a significantly larger population of MDV sized objects in PFN1 KO cells (Fig. 4C,D). To qualify as an MDV, however, a mitochondria fragment also needs to be cargo selective. Hence, we labeled Control and PFN1 KO cells with antibodies against TOM20 and the mitochondria matrix protein PDH and showed that most mitochondria fragments that met the MDV size criteria were also single cargo vesicles and that PFN1 KO cells had significantly more

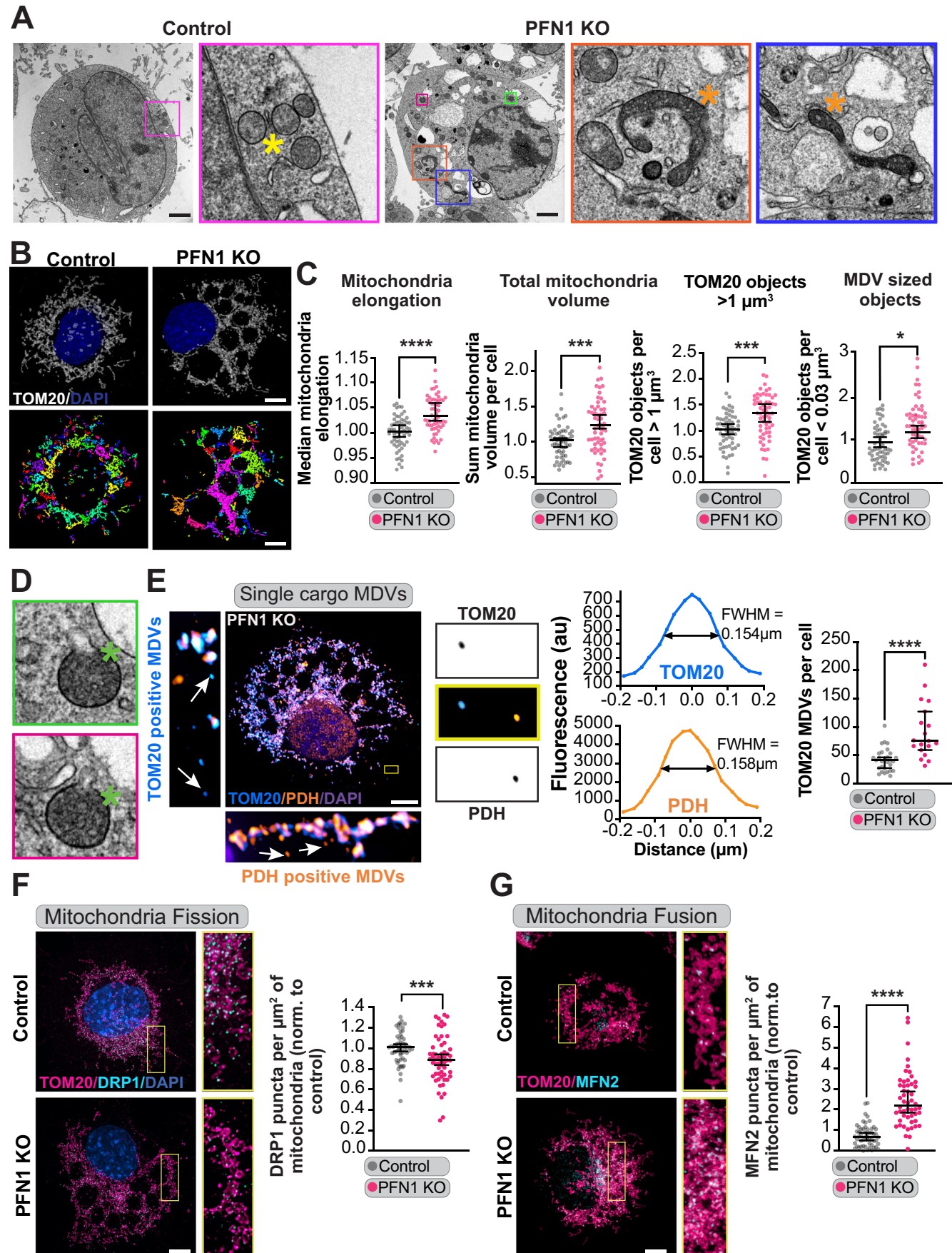

**Figure 4. Loss of PFN1 disrupts mitochondrial morphology.**

(A) Representative transmission electron micrographs of Control and PFN1 KO cells. Control cells have normally shaped mitochondria (yellow stars) whereas PFN1 KO cells have elongated, dysmorphic mitochondria (orange stars). Scale bar = 2 μm. (B) Representative images of TOM20 labeled mitochondria and DAPI in Control and PFN1 KO cells (top) and the results of TOM20 segmentation (bottom). Segmented TOM20 objects were assigned a random color. Note, these images are maximum intensity projections from confocal z-stacks made for presentation. TOM20 segmentation was performed in 3D from the z-stack. Scale bar = 10 μm. (C) Quantification of median mitochondria elongation, total mitochondria volume, the number of TOM20 objects >1 μm$^3$ per cell, and the number of mitochondria-derived vesicle (MDV) sized objects (TOM20 objects <0.03 μm$^3$) per cell from (C). Mitochondria elongation is measured by the length of the longest axis of a 3D object divided by the average of the two smaller axes. Each data point represents one cell, in the case of mitochondria elongation, a data point represents the median elongation value of all mitochondria measured from one cell ($n = 59$ cells for control and $n = 60$ for PFN1 KO). Data is shown as median ± 95% CI. Significance was calculated using a Student's t-test except for MDV sized objects, which used a Mann–Whitney test. (D) Representative transmission electron micrographs of a PFN1 KO cell with MDVs budding from mitochondria (green stars). These images are from the same cell shown in (A). (E) Representative image of a PFN1 KO cell stained for TOM20 and PDH (left). The white arrows on the enlarged insets point to MDV-sized puncta containing only TOM20 or PDH. Quantification of the size of a TOM20 and PDH labeled puncta from the yellow bordered inset (middle), where single channel fluorescence images show that each puncta is only positive of one of the two mitochondrial proteins. Full width at half maximum (FWHM) measurements of fluorescence intensity linescans made through the puncta indicate that each is approximately 0.15 μm in circumference. Quantification of the amount of TOM20 labeled MDVs in Control and PFN1 KO cells (right). Each data point on the graph represents the number of TOM20 positive MDVs that were counted in one cell ($n = 30$ for Control and $n = 18$ for PFN1 KO). Data is shown as median ± 95% CI. Significance was calculated using a Mann–Whitney test. Scale bar = 10 μm. (F) Representative images of Control and PFN1 KO cells stained for TOM20 and the fission protein DRP1 (left). Quantification of DRP1 puncta per μm$^2$ of TOM20 labeled mitochondria normalized to Control (right). Each data point represents the number of DRP1 puncta per μm$^2$ of mitochondria counted in one cell ($n = 60$ for Control and $n = 58$ for PFN1 KO). Data is shown as median ± 95% CI. Significance was calculated using a Mann–Whitney test. Scale bar = 10 μm. (G) Representative images of Control and PFN1 KO cells stained for TOM20 and the fusion protein MFN2 (left). Quantification of MFN2 puncta per μm$^2$ of TOM20 labeled mitochondria normalized to Control. Each data point represents the number of MFN2 puncta per μm$^2$ of mitochondria counted in one cell ($n = 61$ for Control and $n = 60$ for PFN1 KO). Data is shown as median ± 95% CI. Significance was calculated using a Mann–Whitney test. Scale bar = 10 μm. Data information: ****$p < 0.0001$, ***$p < 0.001$, *$p < 0.05$, ns $p > 0.05$. Source data are available online for this figure.

TOM20-only MDVs compared to Control cells (Fig. 4E). The increased production of MDVs further indicates that PFN1 deficient mitochondria are dysfunctional and in need of disposal.

Changes in mitochondria morphology following depletion of PFN1 were not limited to CAD cells. Knocking out PFN1 in mouse embryonic fibroblasts (MEFs) also caused increases in mitochondria size, sum mitochondria volume, and MDV-sized mitochondria fragments (Fig. EV2E,F). The only measured parameter that differed between the two cell types following the loss of PFN1 was elongation, which was substantially reduced in PFN1 KO MEFs (Fig. EV2F,G). Rather than becoming longer, mitochondria in MEFs lacking PFN1 were swollen and more circular (Fig. EV2F). While additional experiments are needed to explain why mitochondria from different cells enlarge in dissimilar ways following the loss of PFN1, it is clear from these experiments that PFN1 is important for controlling mitochondria shape and size.

In response to the metabolic demands of the cell, mitochondria divide, fuse, and undergo directed transport (Liesa and Shirihai, 2013). These activities are necessary for proper mitochondrial function (Adebayo et al, 2021). In addition, mitochondrial dynamics and mitophagy are closely linked (Kleele et al, 2021). To assess mitochondrial mobility and dynamics in the absence of PFN1 we performed live cell imaging of control and PFN1 KO cells. Every mitochondrion in the cell was imaged using a spinning disk confocal microscopy protocol that allowed fast volumetric sampling without compromising spatial resolution (detailed in Methods). Mitochondria were then automatically segmented and tracked using MitoMeter software (Lefebvre et al, 2021). With this approach, we found that there was no significant difference in mitochondria velocity or distance traveled between Control and PFN1 KO cells (Fig. EV2B), indicating that observed mitochondria defects were not caused by a loss of mobility. To investigate whether increased elongation and mitochondria volume could be connected to differences in mitochondria fission/fusion events, we stained cells for DRP1, an important fission protein (Korobova et al, 2013), and MFN2, which governs mitochondria OMM fusion

(Chen et al, 2003). We found that DRP1 localization to mitochondria was decreased, and MFN2 was strongly increased (Fig. 4F,G) in PFN1 KO cells, which may explain why mitochondria are larger in the absence of PFN1.

## PFN1 localizes inside of mitochondria

The most obvious explanation for PFN1's importance in maintaining the functional integrity of mitochondria would be that it facilitates actin polymerization on the mitochondria surface required for fission. However, depolymerizing a substantial amount of cytosolic actin does not substantially depolarize mitochondria (Fig. 3H) or trigger mitophagy (Fig. 2F). While it could be argued that the specific type of actin filaments involved in mitochondrial dynamics are not affected by the dose of Latrunculin A used in those experiments, another possibility is that PFN1 has some other mitochondrial specific function that has yet to be described. Notably, both actin and myosin have been identified in mitochondria fractions (Dadsena et al, 2021; Ohnishi and Ohnishi, 1962; Reyes et al, 2011; Takahashi et al, 2018; Xie et al, 2018) and it has been suggested that actin can regulate mitochondrial morphology and respiratory properties from within (Xie et al, 2018). In addition, since swelling agents have non-uniform effects on mitochondria morphology (Zorov et al, 2019), it has been speculated that cytoskeletal elements within mitochondria provide localized structure and force generation. However, PFN1 has not previously been detected in any of the spatial proteomic screens for mitochondrial proteins (Callegari et al, 2019; Chen et al, 2019; Cho et al, 2020; Cole et al, 2015; Han et al, 2017; Han et al, 2019; Hung et al, 2017; Hung et al, 2014; Janer et al, 2016; Kwak et al, 2020; Lam et al, 2015; Rhee et al, 2013; Yoshinaka et al, 2019).

Although PFN1 has been shown to associate with microtubules (Nejedla et al, 2021; Nejedla et al, 2016; Pimm et al, 2022) and stress granules (Figley et al, 2014), its high concentration and uniform cytoplasmic distribution often confound studies trying to identify its subcellular localization (Jiang et al, 2022; Pimm et al, 2022). Therefore,

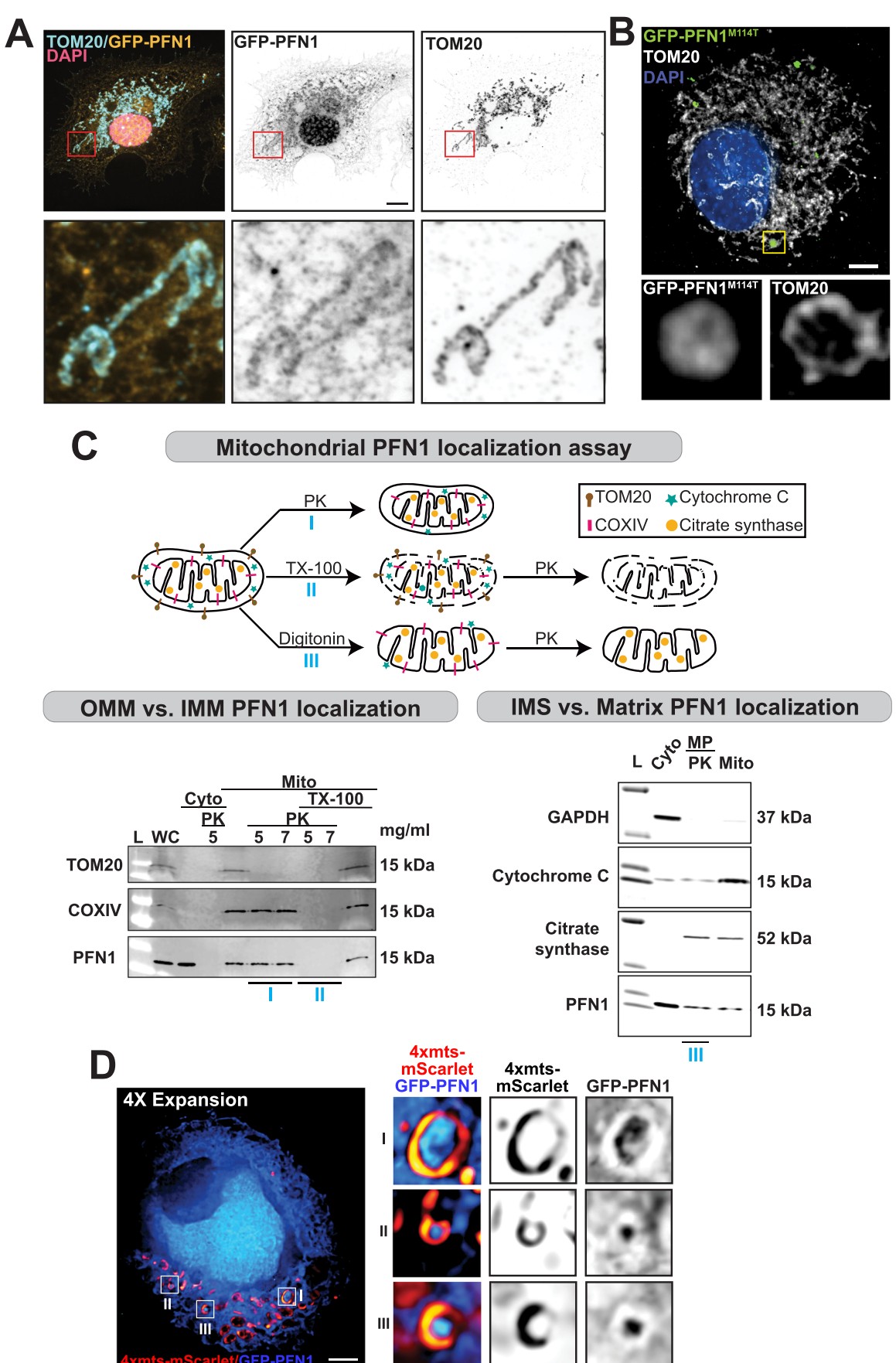

◀  **Figure 5.  PFN1 is inside of mitochondria.**

(A) An image showing TOM20, GFP-PFN1, and DAPI in a PFN1 KO cell that was subjected to an immunocytochemistry protocol that had increased detergent extraction after fixation to remove cytosolic GFP-PFN1. Scale bar = 10 μm. (B) An image showing TOM20, the ALS mutant GFP-PFN1^M114T, and DAPI in a PFN1 KO cell. The inset shows a mitochondrion where the GFP-PFN1^M114T has aggregated inside of the TOM20 labeled membrane. Scale bar = 10 μm. (C) (Top) Schematic protocol of experimental design to determine mitochondrial localization of PFN1. Mitochondria were isolated from Control CAD cells and then either treated with Proteinase K (PK) to digest all the proteins outside of the outer mitochondrial membrane (OMM) (I), treated with the detergent Triton X-100 (TX-100) and then Proteinase K to digest all of the proteins inside the OMM (II), or treated with the detergent Digitonin to remove the OMM and make mitoplasts (MP) and then PK to digest the proteins outside of the inner mitochondrial membrane (IMM) (III). PFN1 was assessed along with the cytoplasmic protein GAPDH, the OMM protein TOM20, the intermembrane space protein Cytochrome C, the IMM protein COXIV, and the matrix protein Citrate synthase. (Bottom left) Western blot of TOM20, COXIV, and PFN1 from experiments I and II to determine if PFN1 is located inside the OMM. PFN1 is protected from Proteinase K unless mitochondria are pre-treated with Triton X-100, like the IMM protein COXIV but not the OMM protein TOM20. Also shown is the cytoplasmic fraction (Cyto) with and without Proteinase K treatment, demonstrating that the amount of Proteinase K used in these experiments is enough to digest all the PFN1 from the cytoplasmic fraction. (Bottom right) Western Blot of GAPDH, Cytochrome C, Citrate Synthase, and PFN1 from experiment III to determine if PFN1 is located in the mitochondrial matrix. PFN1 is protected in mitoplasts treated Proteinase K, like the matrix protein Citrate synthase but not the intermembrane space protein Cytochrome C. (D) A maximum intensity projection image of a PFN1 KO cell expressing 4Xmts-mScarlet to label the IMM and GFP-PFN1 from a 4X expansion microscopy experiment (left). The zoomed insets (right) show different focal planes from the confocal z-stack where GFP-PFN1 was found inside of the IMM. Scale bar = 10 μm. Source data are available online for this figure.

to determine if PFN1 was localized to mitochondria, we increased the concentration of detergent in our immunocytochemistry protocol to enhance the post-fixation extraction of cytoplasmic components. With this approach, we found that GFP-PFN1 clearly colocalized with TOM20 labeled mitochondria (Fig. 5A). In addition, ALS-linked GFP-PFN1 mutants that form insoluble protein inclusions (Wu et al, 2012) were found to occasionally aggregate inside of the TOM20 labeled OMM, causing mitochondria to swell and enlarge (Fig. 5B).

We next sought to determine if endogenous PFN1 could also be found inside mitochondria. We purified mitochondria from CAD cells using the nitrogen cavitation technique, which disrupts the plasma membrane without subjecting mitochondria to shear stress, leaving them intact and functional (Gottlieb and Adachi, 2000). After titrating Proteinase K to find the most effective concentration (Fig. EV3A), we divided the mitochondria fraction into two groups (Fig. 5C). One was treated with Proteinase K to digest all the proteins attached to the outer surface of the mitochondria. The second group was treated with Triton X-100 (TX-100) to permeabilize the mitochondria prior to Proteinase K treatment, allowing the digestion of proteins in the inner mitochondrial membrane and the matrix. As shown in Fig. 5C, treatment with Proteinase K alone completely removes the OMM protein TOM20 from the mitochondria fraction but does not remove the IMM protein COXIV, or PFN1. Pre-treatment with TX-100 followed by Proteinase K successfully removes both the IMM protein COXIV and PFN1 from the mitochondria fraction. These data show that PFN1 is inside of the OMM and not on the mitochondrial surface.

To further explore PFN1's internal localization in mitochondria, we removed the OMM from isolated mitochondria with Digitonin treatment to generate mitoplasts (Sileikyte et al, 2011; van Vlies et al, 2007) (Fig. EV3B). A portion of mitoplasts were treated with Proteinase K to digest all the proteins attached to the surface of the remaining IMM, followed by western blotting. Proteinase K treatment of mitoplasts reduced the intermembrane space (IMS) component cytochrome C but not the matrix protein citrate synthase or PFN1, strongly suggesting that PFN1 is primarily in the mitochondrial matrix. Further evidence for the matrix localization of PFN1 was obtained using 4X expansion microscopy (Chen et al, 2015) to enhance the spatial resolution of cells expressing GFP-PFN1 and an IMS localized 4xmts-mScarlet. We were able to identify multiple examples of GFP-PFN1 clearly inside of a closed

IMS-labeled boundary (Fig. 5D). Finally, following the publication of our preprint (Read et al, 2023), another group found mitochondria localized PFN1 in human embryonic kidney cells via cross-link assisted spatial proteomics (Zhu et al, 2024). However, their work suggested that PFN1 is present in both the IMS and the matrix. Although our data indicates a matrix-specific localization for PFN1 (Fig. 5C), further experiments will be needed to conclusively differentiate between a matrix versus IMS localization, and to determine whether its localization might change in response to the metabolic or dynamic demands of the mitochondria.

In summary, we have shown that the actin-monomer binding protein PFN1 is critical for mitochondrial homeostasis. Without PFN1, mitochondria become large and elongated (Fig. 4A–C), have altered dynamics (Fig. 4E,F), are depolarized, and have increased superoxide production (Fig. 3G–I). Furthermore, cells lacking PFN1 are defective in glycolysis (Fig. 3B,C). The accumulation of dysfunctional, dysmorphic mitochondria in PFN1 KO cells activates quality control mechanisms such as mitophagy (Fig. 2A–C) and the production of MVDs (Fig. 4C–E). ALS-linked PFN1 mutants were not able to compensate for PFN1 loss of function effects on mitochondria (Fig. EV2D) and can aggregate inside the OMM (Fig. 5B). Since size and number of PFN1 aggregates are known to increase with disease progression (Yang et al, 2016), a combination of loss and gain of toxic function in mitochondria could explain how mutant PFN1 causes cellular toxicity and drives neurodegeneration in ALS.

Finally, we have demonstrated that mitochondria depolarization and initiation of mitophagy are not caused by the loss of cytoplasmic actin filaments (Figs. 3H and 2F), but rather due to a specific and novel role for PFN1 in maintaining mitochondrial integrity. In support of this, we found that PFN1 was localized inside mitochondria (Fig. 5C,D). We have yet to determine the exact role PFN1 has inside mitochondria, given the difficulty in separating its cytoplasmic function from potential novel activities inside mitochondria. These challenges include separating out indirect effects on mitochondria by proteins that bind to actin filaments in the cytosol (DeWane et al, 2021; Garcia-Bartolome et al, 2017), or by actin polymerization at the mitochondria surface, which is also linked to respiration (Chakrabarti et al, 2018). However, since mitochondria are a rigorously gated cellular compartment (Hoogewijs et al, 2018; Szczepanowska and

Trifunovic, 2022), it is unlikely that PFN1 would be there without serving an essential purpose.

# Methods

## Cell lines and cell culture

### Cath.a-differentiated (CAD) cells

Cath.a-differentiated (CAD) cells (Qi et al, 1997) were originally purchased from Sigma-Aldrich. Control and PFN1 KO CAD cells were generated using CRISPR/Cas9 as previously described (Skruber et al, 2020). CAD cells were cultured in DMEM/F12 medium with L-Glutamine (Gibco) supplemented with 8% fetal calf serum (R&D Systems) and 1% penicillin-streptomycin (Corning). Prior to imaging, CAD cells were plated onto coverslips pre-coated with 10 μg/mL laminin from Engelbreth-Holm-Swarm murine sarcoma basement membrane (Sigma-Aldrich) overnight at 4 °C. DMEM/F12 medium without phenol red (Gibco) supplemented with 15 mM HEPES (MP Biomedical) was used for live-cell imaging. To assess autophagic responses, we cultured cells for 6 h in serum- and amino acid-free medium (Krebs–Henseleit medium) which initiates autophagy by starvation. CAD cells differentiate into a neuronal-like cell morphology upon serum withdrawal, but only after 48 h (Qi et al, 1997). We routinely use serum withdrawal to validate CAD cells by ensuring that they can undergo neuronal differentiation as evidenced by the formation of long (>100 μm), narrow projections after 2 days. PFN1 KO cells were regularly checked for PFN1 expression by western blot.

### Mouse embryonic fibroblasts (MEFs)

PFN1 KO MEF clonal lines were established from previously described ARPC2 conditional knockout mice (Rotty et al, 2015). Among these clonal lines, JR20s were used to express Cas9 and our previously verified sgRNA (5'-TCGACAGCCTTATGGCGGAC-3') targeting mouse PFN1 (Skruber et al, 2020) from pLentiCRISPRv2 (Addgene #52961; a gift from Feng Zhang) via lentiviral transduction. Lentivirus was generated through the transfection of pCMV-V-SVG, pRSV-REV, pMDLg/pRRE, and pLentiCRISPRv2-PFN1sgRNA (500 ng each) into HEK293FT cells using X-tremeGENE HP transfection reagent (Roche). Lentivirus was harvested 72 h later and subsequently used to infect JR20 cells supplemented with 4 μg/mL of Polybrene. Roughly 72 h following the addition of lentivirus, JR20s expressing PFN1 sgRNA and Cas9 were selected using 2 μg/mL puromycin treatment for 48 h. A second round of infections was used for the lentiviral expression of Cox8-GFP from a pLentiLox5.0 vector in a similar manner as described above (to label mitochondria). MEF cells were cultured in DMEM medium with 4.5 g/L D-Glucose, L-glutamine, 25 mM HEPES, and without Sodium Pyruvate (Gibco) supplemented with 8% fetal calf serum (R&D Systems) and 1% penicillin-streptomycin (Corning). Prior to imaging, MEFs were plated onto coverslips that had been coated with 10 μg/mL human fibronectin (Corning) for 1 h at room temperature (RT). All cell lines used for this study were routinely tested for mycoplasma using the Universal Detection Kit (ATCC).

## DNA constructs

The following DNA constructs purchased from Addgene were used: EGFP-C1(Plasmid #54759,), mEGFP-PFN1 (Plasmid #56438),

Cox8EGFP-mCherry (Plasmid # 78520), 4xmts-mScarlet-I (Plasmid #98818), 4xmts-mNeonGreen (Plasmid #98876), pLenti-CRISPRv2 (Plasmid #52961) mCherry-Parkin (Plasmid #23956) pMXs-puro GFP-p62 (Plasmid # 38277). GFP-PFN1 and GFP-PFN1[R88E] have been previously described (Skruber et al, 2021). Plasmids expressing the PFN1-ALS mutants *M114T, E117G* and *G118V* were generated by us from the GFP-PFN1 plasmid using site-directed mutagenesis (Q5 New England Biolabs) with the following primers: M114T: GTC CTG CTG ACG GGC AAA GAA G (forward) and CC TTC TTT GCC CGT CAG CAG GAC (reverse); E117G: ATG GGC AAA GGA GGT GTC CAC (forward) and G GAC ACC TCC TTT GCC CAT C (reverse); G118V: ATG GGC AAA GAA GTT GTC CAC GGT GGT TTG (forward) and CAA ACC ACC GTG GAC AAC TTC TTT GCC CAT (reverse). Mutagenesis was confirmed by sequencing (Genewiz). mRuby2-LC3 was generated by subcloning LC3 from pmRFP-LC3 (Plasmid #21075) into the pcDNA3-mRuby2 (Plasmid # 40260) using the 5′ BamHI and the 3′ EcoRI cloning sites. Correct inserts were confirmed by sequencing (Genewiz). All constructs were prepared for transfection using either the GenElute HP Endotoxin-Free Plasmid Maxiprep Kit (Sigma-Aldrich) or NucleoBond Xtra Midi EF kit (MACHEREY-NAGEL).

## DNA electroporation

The Neon Transfection System (Invitrogen) was used to introduce DNA constructs into cells using the 10 μL transfection kit (Skruber et al, 2021). Briefly, cells were grown to of 70–80% confluency, detached using 0.5% Trypsin/EDTA solution (Corning) and pelleted by brief centrifugation. Pellets were rinsed with Dulbecco's phosphate-buffered saline (DPBS, Corning) and resuspended in a minimum amount of buffer R (Invitrogen) with a total of 1 μg of DNA per electroporation reaction (except for the mCherry-Parkin over-expression experiments in which we used 1.2 μg of DNA per electroporation reaction). Thereafter cells were cultured for 14–48 h in antibiotic-free medium prior to additional experimental procedures.

## LC3, DEPTOR, and mTOR western blots

Adherent cells were harvested in RIPA buffer with cOmplete EDTA-free Protease Inhibitor Cocktail Roche (Millipore Sigma), except for probing for p-mTOR, where phosphatase inhibitor (Roche Phospho-Stop, Sigma) was also added. Whole-cell lysates were prepared by membrane disruption using repeated passage through a 27-gauge needle. Protein content was assessed with Pierce BCA Protein Assay Kit (Thermo Fisher) and diluted in SDS buffer stained with Orange G (40% glycerol, 6% SDS, 300 mM Tris HCl, pH 6.8). 10 μg samples were evenly loaded on an SDS-PAGE gel (Novex 4–20% Tris-Glycine Mini Gels, Thermo Fisher, or 15% gel as indicated), except for p-mTOR where 25ug was added. Protein was transferred to a PVDF membrane (0.2 μm, 0.45 μm for mTOR and p-mTOR, both from Immobilon) and blocked in 5% Bovine Serum Albumin (BSA) (Sigma-Aldrich) for 20 min. All antibodies were diluted in 5% BSA and 0.1% Tween-20 (Fisher Scientific). Primary antibodies were incubated at 4 °C overnight and secondary antibodies (Li-Cor; Abcam) were incubated for 2 h at RT. LC3 and GAPDH from whole cell lysate were detected with Li-Cor fluorescent antibodies on an Odyssey detection system (Li-

Cor) or via X-ray film after incubation with a developing reagent (Thermo Fisher), as indicated. WesternSure Pre-Stained Chemiluminescent Protein Ladder (Li-Cor) was used as a molecular weight marker. The following antibodies/dilutions were used: rabbit anti-GAPDH (cat # 2118, 1:3000 dilution, Cell Signaling Technology), rabbit anti-LC3 (cat # 2775, Cell Signaling), rabbit anti-DEPTOR/DEPDC6 (cat # NBP1-49674, Novus biologicals), rabbit anti-mTOR (7C10) (cat # 2983, Cell Signaling) and rabbit-Phospho-mTOR (Ser2481) (cat # 2974, Cell Signaling). For secondary antibodies, goat anti-rabbit Alexa Fluor™ 680 (Li-Cor) was used at 1:3500 dilution for imaging on the Li-Cor Odyssey detection system and goat anti-rabbit HRP (Abcam) was used for X-ray detection. For quantitative westerns, antibody detection was determined to be in the linear range by loading increasing lysate concentration as a function of signal.

## Oxygen consumption and glycolytic rates measurement

The Seahorse XFe96 Extracellular Flux Analyzer was used to measure oxygen consumption rates (OCR) and glycolytic rates of adherent CAD cells in real time. A range of cell seeding densities and a series of titration experiments were initially tested to determine optimal conditions. One day prior to assay, the Seahorse Sensor Cartridge (cat#102416-100, Agilent) was hydrated in Seahorse XF Calibrant (cat#102416-100, Agilent) overnight at 37 °C in a non-$CO_2$ incubator. Utilizing an XF 96 cell culture microplate (cat#102416-100, Agilent), $1 \times 10^4$ CAD cells were plated and maintained in appropriate growth medium. On the day of the assay, cells were examined under the microscope to confirm confluence. Growth medium was gently aspirated, and cells were washed two times with freshly prepared Seahorse XF assay medium (cat#103680-100, Agilent) supplemented with glucose (10 mM), pyruvate (1 mM), and glutamine (2 mM). The cell culture microplate containing the cells was incubated at 37 °C in a non-$CO_2$ incubator for 45–60 min, and then transferred to Seahorse XFe96 Extracellular Flux Analyzer (cat# S7894-10000, Agilent). The sensor cartridge and cell plate were equilibrated and calibrated according to the manufacturer's instructions. To measure OCR, the cells were then subjected to Mito Stress Test (cat#103015-100, Agilent) by sequential injections of mitochondrial inhibitors to constitute final concentration per well of 1 μM Oligomycin, 1 μM Carbonyl cyanide-4 (trifluoromethoxy) phenylhydrazone (FCCP), and 0.5 μM Rotenone/Antimycin A. All dilutions were freshly reconstituted the same day. Analysis was run using the standard Mito Stress assay protocol. Oxygen consumption rates were determined by Seahorse X96 Wave Software, and the data was normalized to protein concentration per well (pmol/min/μg). To measure the glycolytic rate, Seahorse XF Glycolysis Rate Assay Kit (cat#103344-100, Agilent) was used. The same protocol for hydrating cartridge, seeding, and washing cells were used as mentioned above. The glycolytic rate was measured by sequential injections of 0.5 μM Rotenone/Antimycin A and 50 mM of 2-DG. Analysis was run using the standard glycolytic rate assay protocol. Glycolytic rate was determined by Seahorse X96 Wave Software and the data was normalized to protein concentration per well (pmol/min/μg).

## Confocal microscopy

All images were acquired using either a Nikon A1R+ laser scanning confocal microscope or a Nikon CSU-W1 SoRa spinning disk confocal microscope. The A1R+ microscope was equipped with a 1.49 NA 60X Apo TIRF objective, GaAsP detectors, and a piezo stage. To achieve super-resolution using the laser-scanning confocal, zoom settings were used to obtain confocal z-stacks containing images with oversampled pixels (0.03 μm). Subsequent 3D deconvolution (NIS-Elements) resulted in images with approximately 150 nm resolution. The CSU-W1 SoRa spinning disk confocal microscope was equipped with a 100X 1.49NA SR objective, a 60X 1.49 NA Apo TIRF objective, a 40X 1.25 NA SIL silicone oil objective, a 20X 0.75 NA Plan Apo objective, a piezo stage, a Tokai Hit stage top incubator, and a Hamamatsu Fusion BT camera. Images were acquired using either the W1 or SoRa spinning disk modes depending on the needs of the experiment. To achieve super-resolution, confocal z-stacks were acquired using the 100X objective and the SoRa 2.8X magnifier, followed by 3D deconvolution. Background noise was removed from images prior to deconvolution using Denoise.ai, a trained neural network that uses deep learning to estimate and remove the noise component of an image. All image acquisition, processing, and analysis was performed using NIS-Elements software unless otherwise indicated.

## Electron microscopy

Control and PFN1 KO CAD cell pellets were fixed in 4% paraformaldehyde, 2% glutaraldehyde in 0.1 M sodium cacodylate (NaCac) buffer, pH 7.4, postfixed in 2% osmium tetroxide in NaCac, stained en bloc with 2% uranyl acetate, dehydrated with a graded ethanol series and embedded in Epon-Araldite resin. Thin sections were cut with a diamond knife on a Leica EM UC7 ultramicrotome (Leica Microsystems, Inc, Bannockburn, IL), collected on copper grids and stained with uranyl acetate and lead citrate. Cells were observed in a JEM-1400Flash transmission electron microscope (JEOL USA Inc., Peabody, MA) at 120 kV and imaged with a Gatan "OneView" CCD Camera using DigitalMicrograph software (Gatan Inc., Pleasanton, CA).

## Immunofluorescence

Cells were seeded onto coverslips and cultured for at least 2 h prior to fixing and staining. Thereafter cells were fixed with either 4% paraformaldehyde (cat# 15710, Electron Microscopy Sciences) or with 3% PFA + 0.75% Gluteraldehyde (cat# 16019, Electron Microscopy Sciences) for 10 min at RT and then permeabilized for 3 min with 0.2% Triton-X 100 (Millipore Sigma). Over-extracted cells were treated with 0.4% Triton-X 100 for 10 min. Cells were then washed three times with DPBS and stained overnight at 4 °C with primary antibodies diluted in PBS. Next, they were washed twice with PBS for 5 min, incubated with secondary antibodies (diluted 1:1000) for 2 h at RT in PBS. Actin filaments were stained with Alexa Fluor 488, 568 or 647 phalloidin (Life Technologies) for 30 min at RT in DPBS. Cells were washed three times with DPBS before mounting with Prolong Diamond (Life Technologies). The following antibodies were used: Rabbit anti-Tom 20 (cat# 11802-1-AP, Proteintech) 1:300 dilution; rabbit anti-parkin (cat# PA5-13399, Invitrogen) 1:50 dilution, mouse anti-DLP1 (cat# 611113 Clone 8/DLP1 (RUO), BD biosciences) 1:100 dilution, Mouse anti MFN2 (cat# SC-100560, Santa Cruz) 1:500, Rabbit anti PDH E1 (cat# 18068, Proteintech) 1:800. Preabsorbed secondary antibodies used were anti-mouse IgG 647, anti-rabbit

IgG 568 at a 1:1000 dilution. Coverslips were mounted onto slides using ProLong Diamond (Thermo Fisher). All slides were cured at RT in the dark for 2 days prior to imaging.

## Quantification of LC3 puncta

For comparison of LC3 puncta in Control, PFN1 KO, and PFN1 KO + PFN1 conditions, CAD cells were electroporated with plasmids expressing mRuby2-LC3 and GFP or GFP-PFN1. The next day, they were plated onto laminin-coated coverslips for 2 h and then fixed and mounted for imaging. Confocal z-stacks of mRuby2-LC3 were taken using the Nikon A1R+ laser scanning confocal microscope using the 60× objective at settings that met Nyquist criteria. The z-stacks were compressed into a maximum intensity projection, the background was subtracted, and then mRuby2-LC3 was segmented using intensity-based thresholding using size filters (objects needed to be >0.01 μm²). The threshold value was set manually from cell to cell. LC3 puncta were measured by counting the number of segmented puncta per cell. Image acquisition settings were identical for all groups.

For comparison of LC3 puncta in Control and PFN1 KO cells were transfected with mRuby2-LC3 and immediately plated onto laminin-coated coverslips. The next day, they were washed and then treated with new media containing 100 nM Bafilomycin A1, or an equivalent volume of DMSO as a solvent control, for 4 h at 37 °C in a 5% $CO_2$ incubator. They were then fixed and mounted for imaging. Cells were imaged using SoRa spinning disk confocal microscopy using a 100X 1.49NA SR using the 2.8X SoRa magnifier. Confocal z-stacks were denoised using Denoise.ai and deconvolved using the Blind algorithm (20 iterations). Z-stacks were then condensed to a single maximum intensity projection. mRuby2-LC3 puncta were segmented using the Spot Detection function. Identical image acquisition and Spot Detection settings were applied to all experimental groups.

## Quantification of Parkin accumulation at mitochondria

Cells were transfected with mCherry-Parkin and immediately plated onto laminin-coated coverslips. The next day, they were fixed and mounted for imaging. Cells were imaged using SoRa spinning disk confocal microscopy using a 100X 1.49NA SR using the 2.8X SoRa magnifier. Confocal z-stacks were denoised using Denoise.ai and deconvolved using the Blind algorithm (20 iterations). Z-stacks were then condensed to a single maximum intensity projection. Parkin accumulated at mitochondria was segmented through intensity-based thresholding and then the segmented area of each cell was measured. Importantly, identical image acquisition and thresholding settings were applied to Control and PFN1 KO cells.

## Quantification of mitochondria delivery to lysosomes

Cells were transfected with Cox8-EGFP-mCherry and immediately plated onto laminin-coated coverslips. After 24 or 48 h, they were fixed and mounted for imaging. Cells were imaged using SoRa spinning disk confocal microscopy using a 100X 1.49NA SR using the 2.8X SoRa magnifier. Identical image acquisition and thresholding settings were applied to all experimental groups. At least 20 random fields of view were selected for each condition for each

biological replicate. Confocal z-stacks were denoised using Denoise.ai. Z-stacks were then condensed to a single maximum intensity projection. Due to the variance in background fluorescence between cells, mCherry-only positive puncta were manually counted. The criteria selecting puncta was a lack of GFP fluorescence and an mCherry fluorescence that was 2:1 over background, measured by subtracting the background from the fluorescence peak of a line drawn through the puncta using the Intensity Line Profile tool.

To determine the effects of depolymerized actin on mitochondria delivery to lysosomes, wild-type CAD cells were transfected with the Cox8-EGFP-mCherry plasmid, and cultured on laminin-coated coverslips for 48 h, then treated with 10–20 nM Latrunculin A dissolved in DMSO (Sigma) or an equivalent volume of DMSO as control. Cells were treated for 14 h at 37 °C in a standard tissue culture incubator, before being fixed and stained with phalloidin to confirm actin depolymerization. Image acquisition and analysis was performed as described above.

## Quantification of mitochondrial membrane potential

Mitochondrial depolarization was measured using the TMRE-Mitochondrial Membrane Potential Assay Kit (cat# ab113852, Abcam) following manufacturer's instructions. Briefly, 5 nM TMRE dye diluted in cell culture medium was added to cells plated on coverslips and then incubated for 30 min at 37 °C. Thereafter the dye was removed, the cells were washed once in warm imaging medium, prior to mounting the coverslips in live cell imaging chambers and imaged at 20X with the Nikon CSU-W1 SoRa spinning disk confocal microscope. Identical image acquisition and thresholding settings were applied to all experimental groups.

To determine the effects of depolymerized actin on mitochondrial membrane potential, we cultured WT CAD cells (control cells) on laminin-coated coverslips for 6 h before treating them with 10 nM Latrunculin A dissolved in DMSO (Sigma). We have previously shown that low dose overnight treatment with Latrunculin A (10–20 nM), reduces the amount of polymerized actin by about 40%, without comprising cell morphology or survival (Cisterna et al, 2024). Cells were treated for 14 h at 37 °C in a standard tissue culture incubator, before proceeding with TMRE staining (as described above) and mounting the coverslips for live cell imaging and analysis.

## Quantification of mitochondrial superoxide production

Mitochondrial superoxide production was measured using the Mitosox Green probe, which produces bright green fluorescence upon oxidation by mitochondrial superoxide (cat# M36006, Thermo Fisher). Briefly, cells were electroporated with 4xmts-Scarlet plasmid to label mitochondria, plated directly onto coverslips and cultured overnight. The next day, following manufacturer's instructions, cells were incubated with 1 μM MitoSOX green in cell culture medium at 37 °C for 30 min, after which they were washed and mounted in live cell imaging chambers and imaged at 20X with the Nikon CSU-W1 SoRa spinning disk confocal microscope. To remove superoxide, superoxide dismutase (cat# S5395, Sigma) was added at 1000 units/mL at the same time as MitoSOX green. Identical image acquisition and thresholding settings were applied to all experimental groups.

## Quantification of mitochondria morphology

Cells were plated onto laminin-coated coverslips for 90 min, fixed and stained overnight for TOM20 to label mitochondria. For rescue experiments, cells were transfected 24 h prior. Cells were imaged using the Nikon CSU-W1 SoRa spinning disk confocal microscope with a 100X 1.49NA SR objective using the 2.8X SoRa magnifier. Identical image acquisition settings were applied to all experimental groups. Confocal z-stacks were denoised using Denoise.ai and deconvolved using the Blind algorithm at 20 iterations using NIS-Elements software. This combination of algorithm and number of iterations was chosen because it yielded the highest signal to noise without being destructive to the morphology of TOM20-stained mitochondria. 3D binary objects were created from the deconvolved z-stacks using intensity-based thresholding. The minimum threshold intensity value was manually chosen for each image, and a clean filter was applied to remove punctate background immunofluorescence from the 3D binary mask. To measure mitochondria morphology in PFN1 KO cells rescued with various GFP-PFN1 constructs, cells were transfected and immediately plated onto laminin-coated coverslips. The next day, they were fixed, stained, imaged, and analyzed as described above.

To determine if mitochondria fragments were cargo-selective MDVs, immunocytochemistry against TOM20 and PDH, super-resolution microscopy, and image processing was performed as described above, with the exception that z-stacks were condensed to a single maximum intensity projection. TOM20 and PDH foci were segmented using the Spot Detection function and then counted as MDVs if they were positive for one marker and negative for the other. Out of 12,020 MDV sized objects identified, only 61 were positive for both Tom20 and PDH. Identical image acquisition and Spot Detection settings were applied to all experimental groups.

## Quantification of Drp1 and Mfn2 localization on mitochondria

Immunocytochemistry against Drp1 and TOM20, or Mfn2 and TOM20 was performed as described above. Cells were imaged using SoRa spinning disk confocal microscopy using a 100X 1.49NA SR using the 2.8X SoRa magnifier. All image processing and analysis described below were performed using NIS-Elements software. Confocal z-stacks were denoised using Denoise.ai and deconvolved using the Blind algorithm (20 iterations). Z-stacks were then condensed to a single maximum intensity projection. TOM20 labeled mitochondria were segmented using intensity-based thresholding and puncta of Drp1 and Mfn2 were segmented using the Spot Detection function. Then, the number of Drp1 or Mfn2 puncta on mitochondria was counted by identifying points where the two segmented features overlapped. The same image acquisition, spot detection, and intensity thresholding parameters were applied to all experimental groups.

## Live cell imaging of mitochondria dynamics

### Live cell microscopy and image processing

Live imaging of mitochondria was performed using the Nikon CSU-W1 SoRa spinning disk confocal microscope. Confocal z-stacks were made using the 40X SIL silicone oil objective, the 4.0X SoRa magnifier, and 2 × 2 camera binning. Image exposure times were ≤10 ms so that entire cell volumes could be imaged every 2–5 s. Background noise was removed using Denoise.ai and images were deconvolved using the 3D Blind Deconvolution algorithm (10 iteration). Before analysis, images were exported into Fiji/ImageJ and corrected for photobleaching using the histogram matching algorithm.

### Quantification of mitochondria dynamics

Analysis of mitochondria dynamics from live cell imaging experiments was performed using a modified version of the Mitometer pipeline (Lefebvre et al, 2021). The segmentation of images was achieved by employing a Frangi filter to enhance the visualization of mitochondria (Frangi et al, 1998) a similar approach as MitoGraph (Viana et al, 2015). During Frangi filtration, the Hessian matrix was thresholded at the square root of the maximum of the Frobenius norm for computational efficiency. Semantic segmentation was achieved via a threshold of 1e−05, where anything in the Frangi-filtered image above the threshold was kept as a mitochondrial pixel. Subsequently, instance segmentation of individual mitochondria was carried out using connected components derived from the semantically segmented image. Skeletonization of the image was performed using Lee's method (Lee et al, 1994) implemented via scikit-image. Instance segmentation of mitochondrial branches was then carried out by traversal between branch junctions and branch tips, or branch tips and branch tips. Mitochondrial tracking was performed as detailed in Mitometer. All parameters were kept constant for analysis of all images to ensure comparable results between samples.

## Expansion microscopy

Sample preparation for 4X expansion microscopy was performed using methods and reagents previously published (Zhang et al, 2020), with some minor adjustments. Cells were electroporated with GFP-PFN1 and 4xmts-mScarlet (1 μg DNA total) and cultured overnight. The next day, cells were seeded onto 13-mm-diameter round coverslips (cat# 174950, Thermo Fisher) pre-coated with 10 μg/mL laminin (Sigma) and allowed to adhere for 4 h at 37 °C in a standard tissue culture incubator. Cells were then fixed with 4% paraformaldehyde (Electron Microscopy Sciences) at RT for 30 min, washed with DPBS (Corning), and post-fixed with 0.25% Glutaraldehyde (Electron Microscopy Sciences) for 10 min followed by a 10-min quenching step (100 mM Glycine, Sigma). Next, cells were washed with DPBS and overlaid with 250 μL of gelation solution (Zhang et al, 2020) and placed on ice for 5 min. Thereafter, coverslips with cells were removed and placed on prepared spacer slides (Zhang et al, 2020) with cells facing downwards. 40 μL of gelation solution was then pipetted between the spacer slides and coverslip and allow to gel completely in a humidity chamber at 37 °C for 1 h. Once the gel had formed, coverslips with gels were removed from the chamber and cooled at RT for 2 min. Next, coverslips with gels were placed in a 10 cm tissue culture dish (Corning), gel facing up, covered with aluminum foil, and incubated for 3 h at RT in Proteinase K digestion buffer (Zhang et al, 2020) on an orbital rocker at 60 rpm. After digestion, gels which had now detached from coverslips were removed and placed in a clean 10 cm dish, and expanded by 3 serial incubations in H₂O on an orbital rocker for 60 min at RT. Following complete expansion, the gels were transferred to a 6 cm glass bottom dish pre-coated with 0.1% (weight/volume) Poly-D-lysine (Sigma) for imaging.

## Assay to determine protein localization to the mitochondrial matrix

### Purification of crude mitochondrial fractions

Adherent cells were grown to 100% confluence in DMEM/F12 medium supplemented with 10% fetal bovine serum and 1% penicillin-streptomycin. Cells were washed twice with DPBS and harvested in Mitochondria Isolation Buffer (MIB) (200 mM sucrose, 10 mM Tris-MOPS, 1 mM EGTA-Tris). Next, cells in suspension were held at 150 psi for 20 min on ice in Pierce Protease Inhibitor Cocktail (Thermo Scientific) in a 45 mL Cell Disruption Vessel (Parr Instrument) with a mini magnetic stirrer (Annis et al, 2001). Cells were disrupted by releasing the pressure. Next, the expelled homogenate was pelleted by centrifugation for 3 min (at $300 \times g$ at 4 °C) (Eppendorf). The pellet was discarded, and the supernatant was further separated by centrifugation at $8000 \times g$ for 15 min at 4 °C in a microcentrifuge. The pellet which contained the crude mitochondrial fraction (Mito) was retained, and the supernatant was removed and centrifuged at $20,000 \times g$ for 20 min at 4 °C in a microcentrifuge. The pellet from this final centrifugation step was discarded, and the supernatant containing the Cytosolic fraction (Cyto) was retained.

### Proteinase K/Triton X-100 treatment

The isolated Mito fraction was incubated with 0 or 1% (v/v) Triton X-100 on ice for 3 min. Then, the Mito and Cyto fractions were both incubated with 0, 0.5, 1, 3, 5, or 7 mg/mL proteinase K (Thermo Scientific) dissolved in MIB for 1 h at 37 °C. Next, proteinase K digestions were stopped by 10 mM phenylmethanesulfonyl fluoride (final concentration) (Protease Inhibitor Cocktail, Promega) on ice for 20 min. The samples were then resuspended in RIPA buffer solution (50 mM Tris-HCl pH 7.4, 150 mM NaCl, 1% (v/v) Triton X-100, 0.5% (w/v) sodium deoxycholate, 0.1% (w/v) sodium dodecyl sulfate) with Protease Inhibitor Cocktail (Promega) (van Vlies et al, 2007). Mito and Cyto samples were finally analyzed by western blot.

### Mitoplast generation

The Mito fraction was incubated with 0, 1, 2, or 4 mg Digitonin/mg mitochondrial proteins (Millipore Sigma) at 4 °C for 20 min and agitated at 2000 rpm to eliminate the outer mitochondrial membrane (OMM) and generate Mitoplasts (MPs) (Sileikyte et al, 2011; van Vlies et al, 2007). Digitonin was then removed by pelleting the MPs at $10,000 \times g$ at 4 °C for 10 min, washed with MIB then pelleted again in MIB. Next, the MPs were incubated with 100, 250, or 500 µg/mL proteinase K dissolved in MIB on ice for 10 min to eliminate the external proteins of the MPs. Proteinase K digestions were terminated by adding 10 mM (final concentration) PMSF for 20 min on ice. The samples were then resuspended in RIPA buffer solution with Protease Inhibitor Cocktail. MPs samples were finally analyzed by western blot.

### Western blotting

Whole-cell (WC) lysates were prepared by harvesting with a scraper in RIPA buffer and using repeated passages (5x) through a 21-, 25-, and 27-gauge needle. Mito and MP were prepared by resuspending samples in RIPA buffer with Protease Inhibitor Cocktail and incubated for 10 min at 4 °C at 2000 rpm. Then, protein quantification was assessed with Pierce BCA Protein Assay Kit (Thermo Scientific) and diluted in SDS buffer stained with Orange G (40% glycerol, 6% SDS, 300 mM Tris HCl, 5% β-mercaptoethanol pH 6.8). Next, the samples were denatured at 95 °C for 5 min before loading. 10 µg of samples were loaded on SDS-PAGE gel (Novex 4%–20% Tris-Glycine Mini Gels, Thermo Scientific). Protein was transferred to a PVDF membrane 0.2 µm (Amersham) and blocked in 5% Bovine Serum Albumin (BSA) (Sigma-Aldrich) for 20 min. All antibodies were diluted in 5% BSA and 0.1% Tween-20 (Fisher Scientific). The following antibodies/dilutions were used: rabbit anti-COXIV (1:2000 dilution, overnight at 4 °C, cat# ab202554, clone EPR9442(ABC), Abcam); mouse anti-TOMM20 (1:1000 dilution, 2 h at 37 °C, cat# ab56783, clone 4F3, Abcam); mouse anti-TOM20 (1:500 dilution, 2 h at 37 °C, cat#H00009804-M01, clone 4F3, Abnova); rabbit anti-profilin 1 (1:1000 dilution, overnight at 4 °C, cat# ab124904, clone EPR6304, Abcam). rabbit anti-GAPDH (1:1000 dilution, 1 h at RT, cat# 2118S, clone 14C10, Cell Signal); rabbit anti-Cytochrome C (1:1000 dilution, 1 h at RT, cat# ab133504, clone EPR1327, Abcam); rabbit anti-Citrate synthase (1:1000 dilution, 1 h at RT, cat# ab96600, Abcam). For secondary antibodies, goat anti-rabbit (1:10,000 dilution, 2 h at RT, cat# 926-32211, Li-Cor) and goat anti-mouse (1:10,000 dilution, 2 h at RT, cat# 926-32210, Li-Cor) was used for imaging on the Li-Cor Odyssey detection system.

## Statistical analysis

Unless noted, all quantified data used multiple biological replicates. The Filename Randomizer macro in Fiji/ImageJ was used to perform blinded analysis. Data normalization was done per replicate to account for variability between experiments. Outliers were removed using the ROUT method. All data was tested for normality using the Shapiro–Wilk normality test. If the data assumed a Gaussian distribution, groups were compared using either an unpaired two-sided Student's t test for two conditions or an ordinary one-way ANOVA for three or more conditions. ANOVA was followed by Dunnett's post hoc test for comparisons of all conditions against the control condition or by Tukey's post hoc test for comparisons of all conditions with each other. If the data failed the normality test, then two groups were compared using the Mann–Whitney test and groups of three or more were compared using a Kruskal–Wallis test followed by Dunn's multiple comparisons test. All analysis and graphing of results were performed using Graphpad Prism 10 software.

## Data availability

This study includes no data deposited in external repositories.

The source data of this paper are collected in the following database record: biostudies:S-SCDT-10_1038-S44319-024-00209-3.

## Peer review information

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

## Acknowledgements

We would like to thank Aleksandra Zamaro, Sheela Nagaroti, Masuko Ushio-Fukai, and Tohru Fukai (Augusta University) for assistance with early attempts at purifying mitochondria from CAD cells. Transmission electron microscopy was performed by the Electron Microscopy and Histology Core at Augusta University. Research reported in this publication was supported by the Maximizing Investigators' Research Award from the National Institute of General Medical Sciences of the National Institutes of Health under grant number R35GM137959 to EAV.

## Author contributions

**Tracy-Ann Read**: Conceptualization; Data curation; Formal analysis; Supervision; Investigation; Visualization; Methodology; Writing—original draft; Writing—review and editing. **Bruno A Cisterna**: Formal analysis; Visualization; Methodology; Writing—original draft; Writing—review and editing. **Kristen Skruber**: Formal analysis; Investigation. **Samah Ahmadieh**: Formal analysis; Investigation. **Tatiana M Liu**: Visualization. **Josefine A Vitriol**: Investigation. **Yang Shi**: Formal analysis; Investigation; Visualization. **Joseph B Black**: Resources; Investigation. **Mitchell T Butler**: Resources. **Halli L Lindamood**: Formal analysis; Investigation. **Austin EYT Lefebvre**: Software; Formal analysis. **Alena Cherezova**: Investigation. **Daria V Ilatovskaya**: Investigation. **James E Bear**: Resources. **Neal L Weintraub**: Investigation. **Eric A Vitriol**: Conceptualization; Data curation; Formal analysis; Supervision; Funding acquisition; Investigation; Visualization; Methodology; Writing—original draft; Project administration; Writing—review and editing.

Source data underlying figure panels in this paper may have individual authorship assigned. Where available, figure panel/source data authorship is listed in the following database record: biostudies:S-SCDT-10_1038-S44319-024-00209-3.

## Disclosure and competing interests statement

The authors declare no competing interests.

# Expanded View Figures

**A**

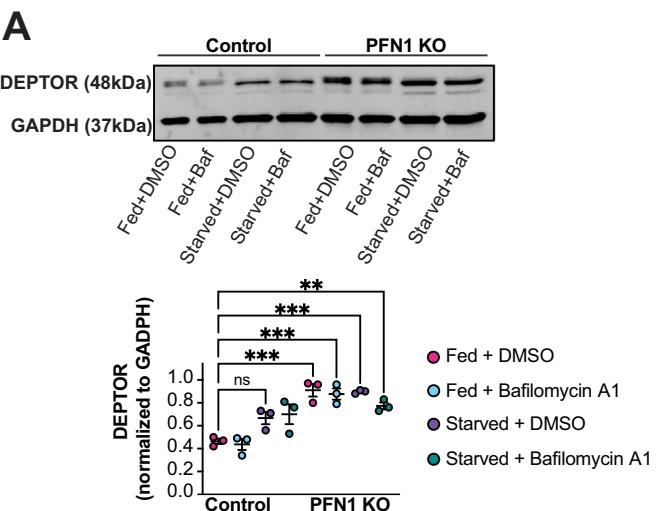

**B**

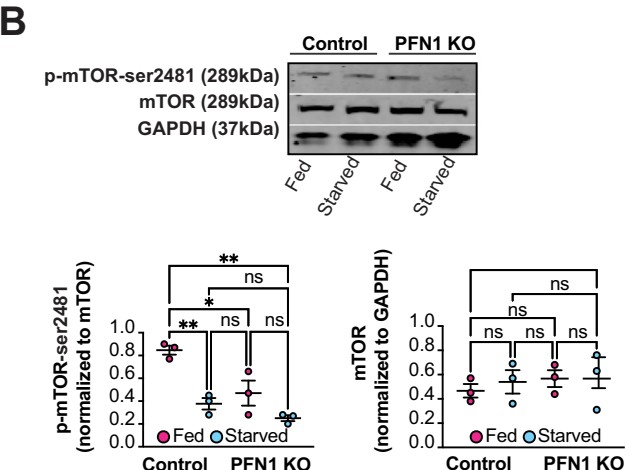

**Figure EV1.   Loss of PFN1 causes an upregulation of autophagy.**

Related to Fig. 1. (**A**) Western blot of DEPTOR and GAPDH in Control and PFN1 KO cells (top). Cells were given normal media (fed) or were nutrient-deprived (starved) for 6 h and were treated with Bafilomycin A (Baf) to inhibit lysosome-mediated degradation or DMSO vehicle control for 4 h. Quantification of DEPTOR normalized against GAPDH (bottom). Data is shown as mean ± SEM and each data point is one biological replicate ($n = 3$). Significance was calculated using ANOVA and Tukey's post hoc test. (**B**) Western blot of mTOR, phospho-S2481-mTOR (p-mTOR-ser2481), and GAPDH in control and PFN1 KO cells (top). Cells were given normal media (fed) or were nutrient-deprived (starved) for 6 h. Quantification of phospho-S2481-mTOR (p-mTOR) (bottom left) and mTOR (bottom right) normalized against GAPDH. Data is shown as mean ± SEM and each data point is one biological replicate ($n = 3$). Significance was calculated using ANOVA and Tukey's post hoc test. Data information: ***$p < 0.001$, **$p < 0.01$, *$p < 0.05$, ns $p > 0.05$. Source data are available online for this figure

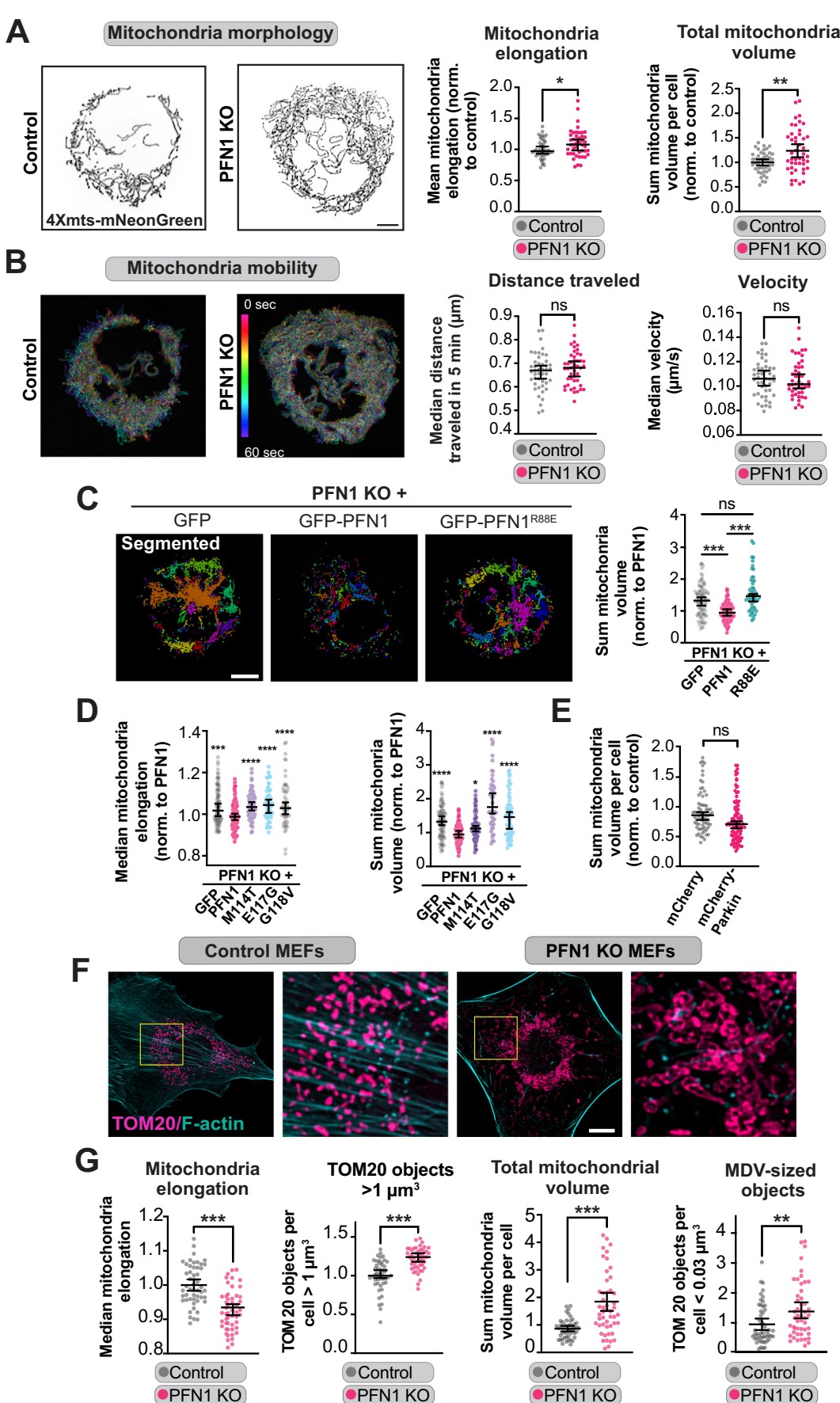

◄ **Figure EV2. Loss of PFN1 disrupts mitochondrial morphology.**

Related to Fig. 4. (**A**) Representative maximum intensity projections of Control and PFN1 KO cells expressing 4xmts-mNeonGreen (left). Quantification (right) of median mitochondria elongation and sum mitochondria volume in Control and PFN1 KO cells from live cell imaging experiments depicted in (**A**). Each data point represents one cell, in the case of mitochondria elongation, a data point represents the median elongation value of all mitochondria measured from one cell. ($n = 45$ cells for Control and PFN1 KO). Data is shown as median ± 95% CI. Significance was calculated using a Student's t test. and time projections from the entire movie (bottom). (**B**) Time projections from live cell imaging of Control and PFN1 KO cells expressing 4xmts-mNeonGreen, color-coded according to the scale inserted in the right image (left). Quantification of median average velocity and median distance traveled of mitochondria (right). Each data point represents the median of all measurements made in one cell. ($n = 45$ cells for Control and PFN1 KO). Significance was calculated using a Student's t test. Scale bar $= 10$ μm. (**C**) Representative images of the segmented TOM20 labeled mitochondria in PFN1 KO cells expressing GFP, GFP-PFN1 or the non-actin binding mutant GFP-PFN1$^{R88E}$ (left). Quantification of sum mitochondria volume. Each data point represents the sum mitochondria volume from one cell ($n = 100$ cells for GFP, $n = 97$ for GFP-PFN1, and $n = 56$ for GFP-PFN1$^{R88E}$). Data is shown as median ± 95% CI. Significance was calculated using a Kruskal–Wallis test followed by Dunn's multiple comparisons test. Scale bar $= 10$ μm. (**D**) Quantification of sum mitochondria and median mitochondria elongation in PFN1 KO cells expressing either GFP, GFP-PFN1, or the ALS-linked mutations GFP-PFN1$^{M114T}$, GFP-PFN1$^{E117G}$ and GFP-PFN1$^{G118V}$. Mitochondria elongation is measured by the length of the longest axis of a 3D object divided by the average of the two smaller axes. Each data point represents one cell, in the case of mitochondria elongation, a data point represents the median elongation value of all mitochondria measured from one cell. ($n = 100$ cells for GFP, $n = 97$ for GFP-PFN1, $n = 77$ for GFP-PFN1$^{M114T}$, $n = 52$ for GFP-PFN1$^{E117G}$, $n = 59$ for GFP-PFN1$^{G118V}$). Data is shown as median ± 95% CI. Significance was calculated using a Kruskal–Wallis test followed by Dunn's multiple comparisons test. (**E**) Quantification of sum mitochondria volume of control cells expressing either mCherry or mCherry-Parkin for 48 h. Each data point represents the sum mitochondria volume from one cell ($n = 75$ for Control and $n = 103$ for PFN1 KO). Data is shown as median ± 95% CI. Significance was calculated using a Mann–Whitney test. (**F**) Representative images of TOM20 labeled mitochondria and F-actin in Control and PFN1 KO cells mouse embryonic fibroblasts (MEFs) (top). Scale bar $= 10$ μm. (**G**) Quantification of median mitochondria elongation, total mitochondria volume, the number of TOM20 objects >1 μm$^3$ per cell, and the number of mitochondria-derived vesicle (MDV) sized objects (TOM20 objects <0.03 μm$^3$) per cell from (**E**). Mitochondria elongation is measured by the length of the longest axis of a 3D object divided by the average of the two smaller axes. Each data point represents one cell, in the case of mitochondria elongation, a data point represents the median elongation value of all mitochondria measured from one cell. ($n = 51$ cells for Control and PFN1 KO). Data is shown as median ± 95% CI. Significance was calculated using a Mann–Whitney test, except for MDV sized objects which used a Student's t-test. Data information: ****$p < 0.0001$, ***$p < 0.001$, **$p < 0.01$, *$p < 0.05$, ns $p > 0.05$. Source data are available online for this figure

**A**

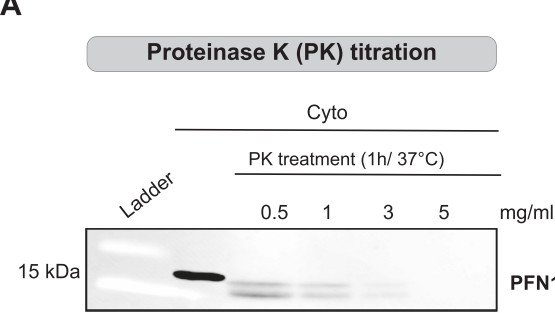

**B**

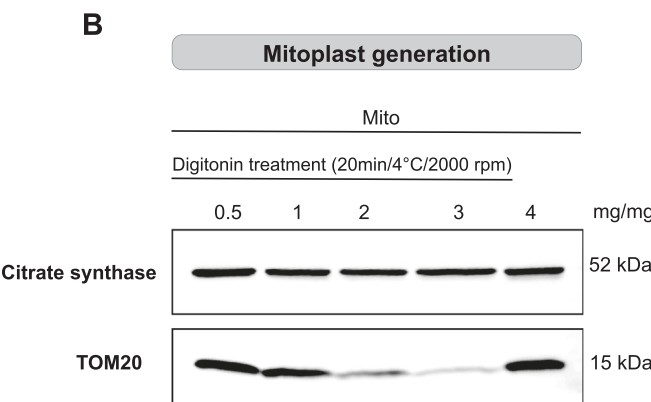

**Figure EV3.   PFN1 is present inside mitochondria.**

Related to Fig. 5. (**A**) Western blot showing titration of Proteinase K applied to the cytoplasmic fraction (Cyto) of CAD cells to identify the optimal concentration at which all cytoplasmic PFN1 is successfully digested. (**B**) Western Blot showing TOM20 (OMM protein) and Citrate synthase (matrix protein) levels from mitochondria treated with increasing amounts of Digitonin to dissolve the OMM to generate mitoplasts. Source data are available online for this figure.

