## [Peer Review File · EMBO Reports]

The actin binding protein profilin 1 localizes inside mitochondria and is critical for their function

Tracy-Ann Read, Bruno Cisterna, Kristen Skruber, Samah Ahmadieh, Josefine Vitriol, Tatiana Liu, Yang Shi, Joseph Black, Mitchell Butler, Halli Lindamood, Austin Lefebvre, Alena Cherezova, Daria Ilatovskaya, James Bear, Neal Weintraub, and Eric Vitriol

Corresponding author(s): Eric Vitriol (evitriol@augusta.edu) , Tracy-Ann Read (tread@augusta.edu)

Review Timeline:

Submission Date:	17th Aug 23
Editorial Decision:	6th Oct 23
Revision Received:	2nd Apr 24
Editorial Decision:	4th Jun 24
Revision Received:	16th Jun 24
Accepted:	28th Jun 24

Editor: Deniz Senyilmaz Tiebe

Transaction Report:

Dear Dr. Vitriol,

Thank you for the submission of your research manuscript to our journal, which was now seen by three referees, whose reports are copied below.

The referees express interest in the proposed role of PFN1 in regulation of mitochondrial function. However, they also raise significant concerns that need to be addressed to consider publication here. In particular,

- The proposed matrix localization of PFN1 requires additional experimental support - e.g. proteinase K assays (referee #1, point 4; referee #2, point 2, referee #3, point starting as 'Figure 6 and manuscript text...')
- The effect of PFN1 depletion on mitochondrial function should be investigated in more depth (referee #3, point starting as 'comments to Figure 3...').
- The effect of PFN1 depletion on the autophagic flux should be better demonstrated (referee #2, points 2, 3, 4).

While we agree with referees #1 and #3 that getting more insight into the mechanism by which PFN1 regulates mitochondria would significantly strengthen the manuscript, not being able to do so experimentally will not preclude from publication. However, all other technical concerns need to be satisfactorily addressed.

Given these positive recommendations, we would like to invite you to submit a revised manuscript. Please revise your manuscript with the understanding that the referee concerns (as in their reports) must be fully addressed and their suggestions taken on board. Please address all referee concerns in a complete point-by-point response. Acceptance of the manuscript will depend on a positive outcome of a second round of review. It is EMBO reports policy to allow a single round of major experimental revision only and acceptance or rejection of the manuscript will therefore depend on the completeness of your responses included in the next, final version of the manuscript.

We realize that it is difficult to revise to a specific deadline. In the interest of protecting the conceptual advance provided by the work, we recommend a revision within 3 months. Please discuss the revision progress ahead of this time with me if you require more time to complete the revisions, or if you have questions or comments regarding the revision (also by video chat).

1. A data availability section providing access to data deposited in public databases is missing (where applicable).
2. Your manuscript contains statistics and error bars based on $n=2$. Please use scatter plots in these cases.

You can submit the revision either as a Scientific Report or as a Research Article. For Scientific Reports, the revised manuscript can contain up to 5 main figures and 5 Expanded View figures, and it should not exceed 27000 characters. If the revision leads to a manuscript with more than 5 main figures it will be published as a Research Article. In this case the Results and Discussion section should be separate. If a Scientific Report is submitted, these sections have to be combined. This will help to shorten the manuscript text by eliminating some redundancy that is inevitable when discussing the same experiments twice. In either case, all materials and methods should be included in the main manuscript file.

2) individual production quality figure files as .eps, .tif, .jpg (one file per figure). See https://wol-prod-cdn.literatumonline.com/pb-assets/embosite/EMBOPress_Figure_Guidelines_061115-1561436025777.pdf for more info on how to prepare your figures.

3) We replaced Supplementary Information with Expanded View (EV) Figures and Tables that are collapsible/expandable online. A maximum of 5 EV Figures can be typeset. EV Figures should be cited as 'Figure EV1, Figure EV2' etc... in the text and their respective legends should be included in the main text after the legends of regular figures.

- Additional Tables/Datasets should be labeled and referred to as Table EV1, Dataset EV1, etc. Legends have to be provided in a separate tab in case of .xls files. Alternatively, the legend can be supplied as a separate text file (README) and zipped

together with the Table/Dataset file.

4) a .docx formatted letter INCLUDING the reviewers' reports and your detailed point-by-point responses to their comments. As part of the EMBO publication's Transparent Editorial Process, EMBO reports publishes online a Review Process File (RPF) to accompany accepted manuscripts. This File will be published in conjunction with your paper and will include the referee reports, your point-by-point response and all pertinent correspondence relating to the manuscript.

<https://www.embopress.org/page/journal/14693178/authorguide#transparentprocess>

5) a complete author checklist, which you can download from our author guidelines

<https://www.embopress.org/page/journal/14693178/authorguide>. Please insert information in the checklist that is also reflected in the manuscript. The completed author checklist will also be part of the RPF.

6) Please note that all corresponding authors are required to supply an ORCID ID for their name upon submission of a revised manuscript (<<https://orcid.org/>>). Please find instructions on how to link your ORCID ID to your account in our manuscript tracking system in our Author guidelines

<<https://www.embopress.org/page/journal/14693178/authorguide#authorshipguidelines>>

7) Before submitting your revision, primary datasets produced in this study need to be deposited in an appropriate public database (see <https://www.embopress.org/page/journal/14693178/authorguide#datadeposition>). Please remember to provide a reviewer password if the datasets are not yet public. The accession numbers and database should be listed in a formal "Data Availability" section placed after Materials & Method (see also

<https://www.embopress.org/page/journal/14693178/authorguide#datadeposition>). Please note that the Data Availability Section is restricted to new primary data that are part of this study. * Note - All links should resolve to a page where the data can be accessed. *

Additional information on source data and instruction on how to label the files are available:

<https://www.embopress.org/page/journal/14693178/authorguide#sourcedata>

9) Our journal encourages inclusion of *data citations in the reference list* to directly cite datasets that were re-used and obtained from public databases. Data citations in the article text are distinct from normal bibliographical citations and should directly link to the database records from which the data can be accessed. In the main text, data citations are formatted as follows: "Data ref: Smith et al, 2001" or "Data ref: NCBI Sequence Read Archive PRJNA342805, 2017". In the Reference list, data citations must be labeled with "[DATASET]". A data reference must provide the database name, accession number/identifiers and a resolvable link to the landing page from which the data can be accessed at the end of the reference. Further instructions are available at <http://www.embopress.org/page/journal/14693178/authorguide#referencesformat>

10) Regarding data quantification (see Figure Legends:

<https://www.embopress.org/page/journal/14693178/authorguide#figureformat>)

- the name of the statistical test used to generate error bars and P values,

- the number (n) of independent experiments (please specify technical or biological replicates) underlying each data point,

- the nature of the bars and error bars (s.d., s.e.m.),

- If the data are obtained from n Program fragment delivered error ``Can't locate object method "less" via package "than" (perhaps you forgot to load "than"?) at //ejpvfs23/sites23b/embor_www/letters/embor_decision_revise_and_review.txt line 56.' 2, use scatter blots showing the individual data points.

12) Please also note our reference format:

I look forward to seeing a revised version of your manuscript when it is ready. Please let me know if you have questions or comments regarding the revision.

Kind regards,

Deniz Senyilmaz Tiebe

Deniz Senyilmaz Tiebe, PhD
Editor
EMBO Reports

Referee #1:

In this manuscript by Read et al., the authors propose that the actin-related protein profilin 1 (PFN1) affects mitochondrial function directly, independently of its role in actin cytoskeleton regulation. These data extend the known function of profilin 1 in mitochondrial homeostasis to the direct control of mitochondrial function and move the field into mitochondria-centric research. Specifically, they tested the activation of mitophagy upon treatment with the actin-depolarizing latrunculin and found no changes. On the other hand, deficiency of PFN1 leads to a significant increase in mitophagy. The authors also show an accumulation of dysfunctional mitochondria in PFN1-deficient cells. Interestingly, mitophagy appears to be insufficient to protect PFN1-deficient cells from the expansion of defective mitochondria.

The actin-affecting PFN1 mutations, including those linked to ALS, failed to rescue mitochondrial defects also detected in PFN1-deficient cells. Additional studies (in preparation) suggest that mutants with a complete or partial loss of PFN1 function in actin polymerization cannot rescue the mitochondrial defects, suggesting that not actin polymerization level per se, but deficiencies in other pathways could lead to mitochondrial decline. Thus, the work provides new evidence on PFN1's role in disease. Consistent with the mitochondria-specific role of PFN1, a subset of this protein appears to localize inside of the mitochondria, and ALS-linked mutant forms intramitochondrial aggregates that are considered to be toxic.

This work is interesting, and the manuscript is easy to follow. The experiments are technically sound, and data mainly support the notion that PFN1 has a direct mitochondrial role independent of the extramitochondrial actin cytoskeleton. However, the mechanism by which PFN1 deficiency induces mitochondrial toxicity, and therefore the mitochondrial function of this protein, has not been evaluated. Furthermore, some important technical and conceptual issues should be addressed before this work can be recommended for publication.

Specific comments:

1. The authors state that 10-20nM latrunculin leads to a 40-50% decrease in polymerized actin. No data is supporting this statement. The images of mitochondria in Figure 2H should also contain actin and actin quantification graphs. Furthermore, latrunculin is likely to have a much more global effect on actin cytoskeleton than depletion of PFN1. Therefore, profilin's "toxic" effects could manifest in the presence of the profilin 1-independent actin cytoskeleton (such as in PFN1 ko cells) but disappear when the whole actin cytoskeleton is affected (such as upon treatment with latrunculin). Can the authors also show clear actin staining (not only cortical but also intracellular) and actin quantifications in PFN1 ko cells? Furthermore, experiments with latrunculin treatment of PFN1 ko cells could give more insights into this issue.
2. Does latrunculin treatment affect mitochondrial activity? The fact that this drug does not induce mitophagy does not preclude the possibility that mitochondrial function is affected. Some steps of mitophagy could require actin cytoskeleton (as shown in Dev Cell. 2018 Feb 26; 44(4): 484-499.e6.) and addressing the possibility that latrunculin-induced dysfunctional mitochondria are not tagged for mitophagy in latrunculin treated cells could strengthen the authors' conclusions.
3. Does Parkin overexpression eliminate mitochondria from PFN1-deficient cells? Suppose the global dysfunction of mitochondria underlies the observed phenotype. In that case, prolonged expression of Parkin should likely lead to the elimination of the whole mitochondrial population in a similar manner as in the case of pharmacologically induced mitochondrial damage. Please address this issue.

4. The mitochondrial localization of PFN1 is perhaps the most surprising and essential discovery described in this manuscript. The data suggest that, indeed, the PFN1 localizes inside the mitochondria. The authors could expand these studies and use osmotic swelling to open the OMM and apply the proteinase K treatment to the swollen mitochondria. Under these conditions, PFN1 should not be significantly degraded only if it localizes in the matrix but would be degraded if present in the intermembrane space or on the outer surface of the IMM. Such data could help in further determination of PFN1's role in the mitochondria. Furthermore, high-resolution images of cells treated with "standard" concentrations of detergent, focusing on the mitochondria and actin, should also be shown in Figure 6. Some quantifications of images in Figure 6A,B is also necessary to inform the reader of the extent of the phenomenon. Other PFN1 mutants should also be included, and their potential mitochondrial localization should be revealed and quantified.
5. The lack of potential mitochondrial pathway candidates regulated by PFN1 is a significant weakness of this work.

Minor:

1. All spliced blots (like in Fig. 1E) should be marked.
2. Segmented images in Figure 4B are flipped vertically. Please correct it.

Referee #2:

The manuscript by Read et al identifies a novel role of PFN1 in the regulation of mitochondria. The authors suggest that PFN1 localizes inside mitochondria where it regulates mitochondrial activity, loss of PFN1 resulting in elongated mitochondria, disrupted glycolysis and increased mitophagy. This is an interesting study that provides novel insights into the roles of PFN1 and the regulation of mitochondrial activity. However, as presented, the data does not fully support the authors' conclusions.

Major points:

1. The authors mention on multiple occasions that they show that PFN1 is localised inside mitochondria, not outside. While their data is consistent with PFN1 being present inside mitochondria, their data does not exclude the presence of extramitochondrial PFN1. In fact, this protein plays important roles that are independent of mitochondria in the cytosol. The authors should thus rephrase their claims. In addition, the expansion microscopy data they used to claim a matrix localisation of PFN1 is not very clear. The PFN1 signal is everywhere (expected) and it is difficult to see whether it is actually enriched within the matrix. Comparison with a known matrix marker would be useful. Alternatively, the proteinase k experiment in Fig. 6C could be expanded to include mitoplasts.
2. While autophagy is affected in the mutants, it is unclear that this actually because autophagy is unregulated as claimed by the authors. For example, the increase in LC3-II caused by bafilomycin seems smaller in PFN1 KO cells (Fig. 1E; difficult to say as no stats are indicated), which would suggest that autophagic flux is decreased in these cells and thus, lysosomes or transport to lysosomes might be defective. How are LC3 puncta affected by baf?
3. In addition, the data is not completely consistent. Under basal conditions, there are more LC3 puncta in mutant cells, which is rescued by reexpression of PFN1. However, the LC3 quantification in Fig. 1E does not completely reflect this (there is even more LC3-II in GFP-PFN1 expressing cells -higher autophagy flux?). Also, the differences in p-mTOR described by in the text is only visible upon starvation, which is not mentioned in the text. Thus, the analysis of the autophagy data should be revisited and the phenotypes under basal and starved conditions clearly described.
4. The LC3 puncta are difficult to see in Fig. 1D. Individual channels should be presented.
5. The TEM in figure 2A needs to be quantified.
6. How were the Parkin-GFP images taken in Fig. 2? The way they look, it is difficult to assess Parkin expression or recruitment to mitochondria.
7. On p.4, the authors claim that the data in fig. 2D-E show that "mitochondria are being successfully targeted to lysosomes and degraded in PFN1 KO cells". I agree about the first part, but the data does not show successful degradation. Mitochondria could accumulate in lysosomes because they fail to be degraded. Increased mitophagy should in theory be accompanied by a decrease in total mitochondrial mass, but the data presented in Fig. 4 suggests otherwise. Are there differences in mitochondrial content when assessed by western blot or citrate synthase activity?
8. The 40-50% decrease in actin polymerization occurring following a low dose of LatA needs to be shown within the manuscript, as we do not have access to the manuscript in preparation to assess the claim.
9. The data in fig. 2C and G seems to have been analysed differently. The numbers are different and the bar graph in G does not show individual data points like C. Also, please indicate in every figure legend what individual data points refer to (a cell? An experiment?).
10. I know that the kit that was used to measure ECAR in Fig. 3 is claimed by Agilent to be measuring glycolysis. Nevertheless, the only thing that is measured here is ECAR, not the flux of carbon through glycolysis so, in my opinion, these graphs should be labelled ECAR, not glycolysis. Is there a difference in lactate production between the cells? On a related note, if it is mostly glycolysis (a cytosolic process) that is affected in PFN1 KO cells, how is it a demonstration that mitochondria are metabolically affected in these cells?
11. MDVs are not properly quantified. First, the EM data in Fig. 4A needs to be quantified. Second, MDVs are defined as cargo-selective vesicles. Thus, they need to be identified as small vesicles positive for one mitochondrial marker but not another one. This is clearly defined in the Sugiura review that the authors cite.
12. What does the statement that PFN1 mitochondria are damaged mean? These mitochondria have only a small decrease in OCR and the cells have much less ROS. Is there a difference in mitoSOX or TMRM staining?

13. The methods used to measure mitochondrial dynamics are not super clear. What does the parameter "elongation" actually measure? Is this the equivalent of the aspect ratio? From the methods, fission and fusion were measured automatically based on branch numbers. Does that mean that tip-to-tip fusion and fission (PMID: 36448541) were not considered? Has this method been validated?
14. The authors claim that the changes in mitochondrial morphology caused by PFN1 deletion are nearly identical in the two cell lines they tested, except for length. This is a rather big difference for nearly identical results. More importantly, from the images in Fig. 4B, KO mitochondria seem more swollen and clustered than elongated. It is possible that because mitochondria are all clustered around large vacuoles (at least in this image), the clusters are picked up as one mitochondrion during segmentation.
15. What do the crossbars represent in the graphs? Average \pm SD? SEM? In several figures (obvious in Fig. 4D, F, I, N), the top and bottom bars are not equally distributed above and below what looks like the average. In any case, this needs to be clearly indicated in the figure legend.
16. Intuitively, one would think that the larger the mitochondrial network, the more fusion/fission there would be (assuming a constant number of fission/fusion events/ mitochondrial length). Any explanation as to why this is not the case in control cells? And what does this correlation actually say relative to mitochondrial dysfunction in KO cells? Or is this an issue with the quantification method?
17. Given that DRP1 only partially localises to mitochondria where it forms foci, Pearson's correlation might not be the most sensitive measure. Given the very small difference observed between WT and KO cells, these results should be confirmed by calculating the actual number of DRP1 foci on mitochondria.

Referee #3:

In this work, Read et al. use previously characterized PFN1 KO CAD cells to analyze the role of the actin-binding protein profiling-1 (PFN1) on mitochondrial ultrastructure and dynamics. The authors show that PFN1 loss leads to increased mitophagy and aberrant mitochondrial morphology associated with disturbed mitochondrial dynamics. Based on their findings, they propose the mitochondrial localization of PFN1 within the mitochondrial matrix and a new role for PFN1 in the overall regulation of mitochondrial metabolism and homeostasis. Overall, this work is mostly descriptive and lacks a fundamental molecular explanation connecting the observed findings. My main concern relates to the fact that the only parameter of mitochondrial function analyzed (respiration) is not clearly affected in PFN1 KO cells. I also have several concerns related to data interpretation, a number of points need clarification and some technical aspects of this work must be improved in order to reach the right conclusions.

- Comments to Figure 3: The authors claim unceasingly in the different sections of the manuscript that loss of PFN1 causes functional defects to mitochondria, or that it affects the functional integrity of mitochondria, but the only functional parameter measured relates to mitochondrial respiration (Figures 3A-F). And data interpretation concerning the effects of PFN1 loss on mitochondrial respiration is unconvincing, first because the reduction in mitochondrial basal respiration observed in PFN1 KO cells is very mild, and second because it does not even affect maximum respiration. It is therefore very unlikely that mitochondrial function and metabolism are heavily affected by PFN1 loss. Moreover, if there was a true defect in mitochondrial respiration one would expect to have higher ROS levels instead of decreased ROS, which is what happens in the PFN1 KO cells (Figure 3G). How can these inconsistencies be explained?

The results presented in Figure 3 in fact indicate that PFN1 KO cells display significantly decreased glycolytic rates than control cells, an adaptation occurring in the cytosol and not within mitochondria. And opposite to the authors' claims, glycolytic suppression normally reprograms intracellular energy metabolism towards mitochondrial OXPHOS (increased respiration) in multiple cell types. Maybe it would be an idea to revisit these data by incorporating the results previously obtained from RNAseq experiments regarding the relative expression of the enzymes related to energy metabolism (glycolysis, Krebs, etc), in order to gain some clues about the metabolic adaptations taking place in the KO cells and compare how these results match their functional observations.

- Figure 4C-L: Regarding ultrastructural parameters of mitochondrial morphology, the increase number of elongated mitochondria, number of objects and sum value in PFN1 KO cells could be attributed to an increased mitochondrial mass per cell, as a compensatory adaptation to the decreased glycolytic rates upon PFN1 loss. The authors need to normalize these parameters by mitochondrial mass.

A curiosity, why would the CAD and MEF PFN1 KO cells behave different in terms of mitochondrial morphology?

- Figure 5: Why is there no increased mitochondrial fusion if there is less fission and decreased Drp1 levels? Please analyze other proteins involved in mitochondrial dynamics to have a clear picture of the alterations on mitochondrial dynamics, including Mfn1, Mfn2 or Opa1.

- Figure 6 and manuscript text: Based on the presented data, it remains the possibility that PFN1 localizes attached to the IMM in the mitochondrial intermembrane space (IMS) rather than in the matrix. Several controls are required to convincingly prove the existence of PFN1 in the matrix (WB in Figure 6C): A positive matrix control (mtDNA, any Krebs cycle enzyme, etc), a positive IMS control (cytochrome c) and a negative control for cytosolic contamination.

In addition the manuscript text is confusing, as sometimes the authors refer to the localization of PFN1 in the IMM and sometimes in the matrix.

- General comment: Please add the specific statistical analysis used on each experiment in the figure legends.

Response to Reviewers

We would sincerely like to thank the reviewers for their thorough evaluation of our manuscript. Their expertise was invaluable in helping us design new experiments and analysis to strengthen our conclusions, improve the language we used to describe our findings, and enhance the presentation of our results. We believe this study has been significantly improved through peer review and brought to a more satisfying conclusion. We hope that you agree and now find the manuscript suitable for publication in *EMBO Reports*. Below is our point-by-point response, with our comments written in blue.

Editor:

Thank you for the submission of your research manuscript to our journal, which was now seen by three referees, whose reports are copied below. The referees express interest in the proposed role of PFN1 in regulation of mitochondrial function. However, they also raise significant concerns that need to be addressed to consider publication here. In particular:

- The proposed matrix localization of PFN1 requires additional experimental support - e.g. proteinase K assays (referee #1, point 4; referee #2, point 2, referee #3, point starting as 'Figure 6 and manuscript text...')

We performed the experiment suggested by Reviewer #2 of making mitoplasts and performing new assays with Proteinase K and detergent. Additionally, we performed the western blots for these experiments with the cytosolic contaminant, IMS, and matrix localization controls that were suggested by Reviewer #3. As shown in Fig. 5C, PFN1 is protected from Proteinase K degradation even after the outer membrane was extracted with digitonin, as was the matrix protein citrate synthase. We would like to thank the Reviewers for challenging us to perform this difficult but very convincing matrix localization experiment. Combined with the imaging data showing that GFP-PFN1 localizes inside closed IMM-labeled Fig. 5D, we believe that we have now definitively shown that PFN1 is part of the mitochondria matrix.

- The effect of PFN1 depletion on mitochondrial function should be investigated in more depth (referee #3, point starting as 'comments to Figure 3...').

We have now included data showing that PFN1 KO mitochondria are depolarized (Fig. 3G) and have increased production of superoxide (Fig. 3H)

- The effect of PFN1 depletion on the autophagic flux should be better demonstrated (referee #2, points 2, 3, 4).

We performed a better characterization of autophagic flux as suggested by Reviewer #2. First, we used Bafilomycin and LC3 fluorescence microscopy assay to **A)** verify that PFN1 KO cells have more LC3 puncta and **B)** demonstrate that there is proportionately more LC3 puncta in PFN1 KO cells than Control cells upon Bafilomycin treatment, which is exactly what the previous data showed (Fig. 1B and D). This would argue that there is a similar amount of autophagosome turnover in Control and PFN1 KO cells. We also performed the statistical comparisons on the autophagic flux assay that Reviewer #2 suggested (Fig.1C) and found no difference in the amount of LC3-II in Control, PFN1 KO, or PFN1 KO/Rescue cells in fed or starved conditions when they are treated with Bafilomycin. Again, we interpret this as showing that autophagic flux is comparable in all conditions and that the western blot assay is not sensitive enough to detect the 2-fold change in autophagosomes that are produced in cells lacking PFN1.

While we agree with referees #1 and #3 that getting more insight into the mechanism by which PFN1 regulates mitochondria would significantly strengthen the manuscript, not being able to do so experimentally will not preclude from publication. However, all other technical concerns need to be satisfactorily addressed.

Thank you for this fair assessment. We have addressed any technical concerns below in the individual responses to each comment.

Referee #1:

In this manuscript by Read et al., the authors propose that the actin-related protein profilin 1 (PFN1) affects mitochondrial function directly, independently of its role in actin cytoskeleton regulation. These data extend the known function of profilin 1 in mitochondrial homeostasis to the direct control of mitochondrial function and move the field into mitochondria-centric research. Specifically, they tested the activation of mitophagy upon treatment with the actin-depolarizing latrunculin and found no changes. On the other hand, deficiency of PFN1 leads to a significant increase in mitophagy. The authors also show an accumulation of dysfunctional mitochondria in PFN1-deficient cells. Interestingly, mitophagy appears to be insufficient to protect PFN1-deficient cells from the expansion of defective mitochondria.

The actin-affecting PFN1 mutations, including those linked to ALS, failed to rescue mitochondrial defects also detected in PFN1-deficient cells. Additional studies (in preparation) suggest that mutants with a complete or partial loss of PFN1 function in actin polymerization cannot rescue the mitochondrial defects, suggesting that not actin polymerization level per se, but deficiencies in other pathways could lead to mitochondrial decline. Thus, the work provides new evidence on PFN1's role in disease. Consistent with the mitochondria-specific role of PFN1, a subset of this protein appears to localize inside of the mitochondria, and ALS-linked mutant forms intramitochondrial aggregates that are considered to be toxic.

This work is interesting, and the manuscript is easy to follow. The experiments are technically sound, and data mainly support the notion that PFN1 has a direct mitochondrial role independent of the extramitochondrial actin cytoskeleton.

Thank you for this positive assessment of our work.

However, the mechanism by which PFN1 deficiency induces mitochondrial toxicity, and therefore the mitochondrial function of this protein, has not been evaluated. Furthermore, some important technical and conceptual issues should be addressed before this work can be recommended for publication.

The editor has stated that we do not need to solve the mechanism through which PFN1 is regulating mitochondria to have the paper published in *EMBO Reports* (see above). However, we have addressed the technical and conceptual issues you raised in the individual responses below. Thank you for your thorough evaluation of our manuscript.

Specific comments:

1. The authors state that 10-20nM latrunculin leads to a 40-50% decrease in polymerized actin. No data is supporting this statement. The images of mitochondria in Figure 2H should also contain actin and actin quantification graphs. Furthermore, latrunculin is likely to have a much more global effect on actin cytoskeleton than depletion of PFN1. Therefore, profilin's "toxic" effects could manifest in the presence of the profilin 1-independent actin cytoskeleton (such as in PFN1 ko cells) but disappear when the whole actin cytoskeleton is affected (such as upon treatment with latrunculin). Can the authors also show clear actin staining (not only cortical

but also intracellular) and actin quantifications in PFN1 ko cells? Furthermore, experiments with latrunculin treatment of PFN1 ko cells could give more insights into this issue.

The effect of overnight application of low concentrations of Latrunculin A on actin is detailed in our paper in the *Journal of Cell Biology* which is currently in press (Cisterna et al, *JCB* 2024). We have cited the preprint of that manuscript in this revision and will update that reference when this paper is accepted for publication. The actin phenotype of PFN1 KO cells has been extensively characterized in the manuscript that we first published using these cells (Skruber et al, *Current Biology* 2020) and again in the *JCB* paper mentioned above. This includes Latrunculin A treatment of PFN1 KO cells, which at the low concentrations we use for these experiments, causes no further cytoskeletal changes.

2. Does latrunculin treatment affect mitochondrial activity? The fact that this drug does not induce mitophagy does not preclude the possibility that mitochondrial function is affected. Some steps of mitophagy could require actin cytoskeleton (as shown in *Dev Cell*. 2018 Feb 26; 44(4): 484-499.e6.) and addressing the possibility that latrunculin-induced dysfunctional mitochondria are not tagged for mitophagy in latrunculin treated cells could strengthen the authors' conclusions.

We treated cells overnight with 10 nM Latrunculin A, which is enough to depolymerize 30-40% of the cell's F-actin without altering cell morphology (Cisterna et al, *JCB*, 2024) and measured mitochondria depolarization using TMRE. As shown in Fig. 3G, there was a small but significant depolarizing effect of Latrunculin A, but this effect was nowhere near as strong as the depolarization which is measured in PFN1 KO cells (Fig. 3G). This would argue for a specific role of PFN1 in maintaining mitochondria integrity and preventing them from mitophagic degradation which is separate from its role as an actin polymerizing agent in the cytoplasm. Thank you for suggesting this experiment.

3. Does Parkin overexpression eliminate mitochondria from PFN1-deficient cells? Suppose the global dysfunction of mitochondria underlies the observed phenotype. In that case, prolonged expression of Parkin should likely lead to the elimination of the whole mitochondrial population in a similar manner as in the case of pharmacologically induced mitochondrial damage. Please address this issue.

We overexpressed Parkin in PFN1 KO cells and found no changes to mitochondria volume after 48 hrs when compared to a transfection control (Fig. EV3D), even though Parkin substantially accumulates on mitochondria (Fig. 2B) and colocalizes with p62 (Fig. 2C) after 24 hrs. In light of this, we have modified the language in the manuscript to reflect that while mitophagy is still functioning in PFN1 KO cells, it may not be operating as efficiently as it does in Control conditions and there is likely a late-stage defect occurring after lysosomal acidification. This is also evident by the accumulation of Cox8-GFP-mCherry in acidified lysosomes in PFN1 KO cells, but not controls (compare 24 vs. 48 hrs in Fig. 2D).

4. The mitochondrial localization of PFN1 is perhaps the most surprising and essential discovery described in this manuscript. The data suggest that, indeed, the PFN1 localizes inside the mitochondria. The authors could expand these studies and use osmotic swelling to open the OMM and apply the proteinase K treatment to the swollen mitochondria. Under these conditions, PFN1 should not be significantly degraded only if it localizes in the matrix but would be degraded if present in the intermembrane space or on the outer surface of the IMM. Such data could help in further determination of PFN1's role in the mitochondria. Furthermore, high-resolution images of cells treated with "standard" concentrations of detergent, focusing on the mitochondria and actin, should also be shown in Figure 6. Some quantifications of images in Figure 6A,B is also necessary to inform the reader of the extent of the phenomenon. Other PFN1 mutants should also be included, and their potential mitochondrial localization should be revealed and quantified.

To address the localization of PFN1 to the mitochondria matrix, we performed the experiment suggested by Reviewer #2 of making OMM-lacking mitoplasts with digitonin and performing new assays with Proteinase K. Additionally, we performed the western blots for these experiments using the cytosolic contaminant, IMS, and matrix localization controls that were suggested by Reviewer #3. As shown in Fig. 5C, PFN1 is protected from Proteinase K degradation even after the outer membrane was extracted with digitonin, as was the matrix protein citrate synthase. (Fig. 5C). Regarding the quantification requested, in this figure we are only using the fluorescence microscopy images of GFP-PFN1 simply as additional proof-of-concept examples of the result showing that endogenous PFN1 is in the mitochondria matrix. The presence of ALS-PFN1 aggregates inside of mitochondria is admittedly rare and we have updated the text to reflect that.

5. The lack of potential mitochondrial pathway candidates regulated by PFN1 is a significant weakness of this work.

While we wish that we were able to resolve the mechanism of what PFN1 is doing in mitochondria the experiments required to do this are extremely difficult and will not be able to be performed within the time frame of a manuscript resubmission. We feel that the identification of a functional role of PFN1 in regulating mitochondria and its localization to the matrix is still of sufficient impact to be published in *EMBO Reports*. We look forward to deciphering the mechanism in future studies.

Minor:

1. All spliced blots (like in Fig. 1E) should be marked.

We have modified how spliced blots so that it is obvious that they were spliced. Raw images of the blots will be included in the resubmission of the manuscript as Source Data.

2. Segmented images in Figure 4B are flipped vertically. Please correct it.

We have corrected this image as requested.

Referee #2:

The manuscript by Read *et al* identifies a novel role of PFN1 in the regulation of mitochondria. The authors suggest that PFN1 localizes inside mitochondria where it regulates mitochondrial activity, loss of PFN1 resulting in elongated mitochondria, disrupted glycolysis and increased mitophagy. This is an interesting study that provides novel insights into the roles of PFN1 and the regulation of mitochondrial activity.

Thank you for this positive assessment of our work.

However, as presented, the data does not fully support the authors' conclusions.

In addressing your concerns, we feel that the conclusions made in the original draft of the manuscript have been strengthened. Additionally, we have modified some of the statements made in the previous version in light of new experiments and analysis that have been performed. Thank you for thoroughly reviewing this manuscript and coming up with helpful suggestions to make it better.

Major points:

1. The authors mention on multiple occasions that they show that PFN1 is localised inside mitochondria, not outside. While their data is consistent with PFN1 being present inside mitochondria, their data does not exclude the presence of extramitochondrial PFN1. In fact, this protein plays important roles that are independent of

mitochondria in the cytosol. The authors should thus rephrase their claims. In addition, the expansion microscopy data they used to claim a matrix localisation of PFN1 is not very clear. The PFN1 signal is everywhere (expected) and it is difficult to see whether it is actually enriched within the matrix. Comparison with a known matrix marker would be useful. Alternatively, the proteinase k experiment in Fig. 6C could be expanded to include mitoplasts.

To address the localization of PFN1 to the mitochondria matrix, we made mitoplasts as suggested and performed new assays with Proteinase K and detergent. Additionally, we performed the western blots for these experiments using the cytosolic contaminant, IMS, and matrix localization controls that were suggested by Reviewer #3. As shown in Fig. 5C, PFN1 is protected from Proteinase K degradation even after the outer membrane was extracted with digitonin, as was the matrix protein citrate synthase. Thank you for this suggestion.

We have also redone the expansion microscopy figures to enhance the presentation of the data. This figure now shows merged images and individual channels of three closed IMM compartments containing PFN1 (Fig. 5D).

2. While autophagy is affected in the mutants, it is unclear that this actually because autophagy is unregulated as claimed by the authors. For example, the increase in LC3-II caused by bafilomycin seems smaller in PFN1 KO cells (Fig. 1E; difficult to say as no stats are indicated), which would suggest that autophagic flux is decreased in these cells and thus, lysosomes or transport to lysosomes might be defective. How are LC3 puncta affected by baf?

We performed a better characterization of autophagic flux as suggested by using Bafilomycin and the LC3 fluorescence microscopy assay to **A)** verify that PFN1 KO cells had more LC3 puncta and **B)** demonstrate that there was proportionately more LC3 puncta upon Bafilomycin treatment in PFN1 KO cells than Controls. Both of these points are conveyed in the data (Fig. 1B-D). This would argue that there is a similar amount of autophagosome turnover in Control and PFN1 KO cells.

3. In addition, the data is not completely consistent. Under basal conditions, there are more LC3 puncta in mutant cells, which is rescued by reexpression of PFN1. However, the LC3 quantification in Fig. 1E does not completely reflect this (there is even more LC3-II in GFP-PFN1 expressing cells -higher autophagy flux?). Also, the differences in p-mTOR described by in the text is only visible upon starvation, which is not mentioned in the text. Thus, the analysis of the autophagy data should be revisited and the phenotypes under basal and starved conditions clearly described.

In the autophagic flux western blot assay there is no difference in the amount of LC3-II in Control, PFN1 KO, or PFN1 KO/Rescue cells in fed or starved conditions when they are treated with Bafilomycin (Fig. 1C). Again, we interpret this as showing that autophagic flux is comparable in all conditions and that the western blot assay is not sensitive enough to detect the 2-fold change in autophagosomes that are produced in cells lacking PFN1 .

Regarding p-mTOR, we have added quantification to the western blots in Fig. EV1B. The data show that p-mTOR-ser2481 is as significantly reduced in fed PFN1 KO cells as it is in starved control cells, which is reflective of their chronic activation of autophagy/mitophagy (Fig. 1 and 2).

4. The LC3 puncta are difficult to see in Fig. 1D. Individual channels should be presented.

We have provided the individual channels of LC3 as requested (Fig.1B and D). All images were acquired using identical parameters and have been scaled identically for presentation so that comparisons between the different conditions can be more easily made.

5. The TEM in figure 2A needs to be quantified.

We have quantified mitophagy using both mCherry-Parkin puncta formation on mitochondria (Fig. 2B) and with the Cox8-GFP-mCherry probe (Fig. 2D). The TEM is being used for illustrative purposes.

6. How were the Parkin-GFP images taken in Fig. 2? The way they look, it is difficult to assess Parkin expression or recruitment to mitochondria.

We have redone the mCherry-Parkin experiments using a higher DNA concentration in the transfection. Now the mCherry-Parkin localization in our studies (Fig. 2B) is more similar to what has been previously shown (Narendra et al, JCB, 2008). Additionally, the Parkin puncta seen in PFN1 KO cells is similar to Parkin puncta in FCCP treated control cells (Fig. 2C). Mitochondria depolarization with CCCP/FCCP is a classic experimental paradigm for causing Parkin localization to mitochondria (Narendra et al, JCB, 2008).

7. On p.4, the authors claim that the data in fig. 2D-E show that "mitochondria are being successfully targeted to lysosomes and degraded in PFN1 KO cells". I agree about the first part, but the data does not show successful degradation. Mitochondria could accumulate in lysosomes because they fail to be degraded. Increased mitophagy should in theory be accompanied by a decrease in total mitochondrial mass, but the data presented in Fig. 4 suggests otherwise. Are there differences in mitochondrial content when assessed by western blot or citrate synthase activity?

Revisiting that line, we would agree that it is an overstatement. We have now changed the text to reflect that while mitophagy is still functioning in PFN1 KO cells, it may not be operating as efficiently as it does in Control conditions and there is likely a late-stage defect occurring after lysosomal acidification. This is evident by the accumulation of Cox8-GFP-mCherry labeled mitochondria in acidified lysosomes that occurs in PFN1 KO, but not control cells (compare 24 vs. 48 hrs in Fig. 2D).

8. The 40-50% decrease in actin polymerization occurring following a low dose of LatA needs to be shown within the manuscript, as we do not have access to the manuscript in preparation to assess the claim.

The effect of overnight application of low concentrations of Latrunculin A on actin is detailed in our recently accepted paper in the Journal of Cell Biology (Cisterna et al, JCB, 2024). We have cited the preprint of that manuscript in this revision and will update that reference when this paper is accepted for publication.

9. The data in fig. 2C and G seems to have been analysed differently. The numbers are different and the bar graph in G does not show individual data points like C. Also, please indicate in every figure legend what individual data points refer to (a cell? An experiment?).

Experiments measuring Parkin puncta have been redone as described above and reanalyzed. Since Parkin forms large accumulations on mitochondria (Fig. 2B-C), we have now reported Parkin puncta as the total area of mCherry-Parkin puncta per cell (Fig. 2B) instead of counting individual puncta.

All graphs have now been redone to show individual data points, and the figure legends now explicitly indicate what each data point refers to and what statistics are being reported.

10. I know that the kit that was used to measure ECAR in Fig. 3 is claimed by Agilent to be measuring glycolysis. Nevertheless, the only thing that is measured here is ECAR, not the flux of carbon through glycolysis so, in my opinion, these graphs should be labelled ECAR, not glycolysis. Is there a difference in lactate production between the cells? On a related note, if it is mostly glycolysis (a cytosolic process) that is affected in PFN1 KO cells, how is it a demonstration that mitochondria are metabolically affected in these cells?

The line graph in Fig. 3A is labeled ECAR, the bar graphs in Fig. 3B and C are labeled Basal and Compensatory Glycolysis according to the manufacturers description of the results from this assay. To avoid confusion, we have expanded the text to make it clear to the reader that these are indirect measurements of glycolysis. We have also now provided new data showing that PFN1 KO mitochondria are depolarized (Fig. 3G) and have increased production of superoxide (Fig. 2H), which surely affects their metabolic status.

11. MDVs are not properly quantified. First, the EM data in Fig. 4A needs to be quantified. Second, MDVs are defined as cargo-selective vesicles. Thus, they need to be identified as small vesicles positive for one mitochondrial marker but not another one. This is clearly defined in the Sugiura review that the authors cite.

We performed immunocytochemistry for TOM20 (OMM) and PDH (IMM) to identify vesicles that meet the MDV size requirements and are cargo selective. From a data set of 12020 MDV-sized objects (obtained from 39 cells), only 61 (0.5%) were positive for both TOM20 and PDH. Therefore, the MDV-sized objects we identified in our high content quantification of TOM20 labeled mitochondria (Fig. 4C) should be indicative of the TOM20 positive MDV population. To make sure of this, we quantified only the TOM20 positive MDVs from control and PFN1 KO cells and found that there is a similar increase in these vesicles in PFN1 KO cells relative to controls (compare Fig. 4C to Fig. 4D). However, you are correct in that one doesn't know that a vesicle is an MDV by size alone until cargo selectivity is demonstrated. Therefore, we have re-labeled the title of the graphs in Fig. 4C and Fig. 4EVF and changed the manuscript text to indicate that we are measuring MDV-sized objects to make sure that we are more precisely describing what is being measured.

12. What does the statement that PFN1 mitochondria are damaged mean? These mitochondria have only a small decrease in OCR and the cells have much less ROS. Is there a difference in mitoSOX or TMRM staining?

As suggested, we performed assays using MitoSox Green and TMRE, which showed that PFN1 KO mitochondria have increased production of superoxide (Fig. 3I) and are depolarized (Fig. 3H). The statement that mitochondria in PFN1 KO cells were damaged was derived from the data showing that Parkin was recruited to them and that they were being targeted for degradation by mitophagy. With the new data we have added, we are able to more confidently state this. Thank you for suggesting these experiments.

13. The methods used to measure mitochondrial dynamics are not super clear. What does the parameter "elongation" actually measure? Is this the equivalent of the aspect ratio? From the methods, fission and fusion were measured automatically based on branch numbers. Does that mean that tip-to-tip fusion and fission (PMID: 36448541) were not considered? Has this method been validated?

Elongation is the ratio of the major and minor axis (in 2D) or the ratio between the major axis and an average of the two minor axes (in 3D). This has been clarified in the Material and Methods and Figure Legends. MitoMeter is not capable of measuring tip-to-tip fusion. Additionally, while MitoMeter has been previously validated for measuring fission and fusion (AEYT Lefebvre et al, Nature Methods 2021), we found some inconsistencies in our own results when we were performing validation experiments in response to this concern. Unfortunately, we were not able to resolve this within the time frame of this resubmission. Combined with the issue of not being able to measure tip-to-tip fusion, we have removed the fission/fusion dynamics measurements obtained from MitoMeter from the manuscript. Instead, we performed the more conventional colocalization assays using DRP1 and MFN2 (Fig. 4E-F). These results show reduced DRP1 and increased MFN2 on PFN1 KO mitochondria, indicating less fission and more fusion, which is very consistent with the increased size and elongation of these mitochondria.

14. The authors claim that the changes in mitochondrial morphology caused by PFN1 deletion are nearly identical in the two cell lines they tested, except for length. This is a rather big difference for nearly identical results. More importantly, from the images in Fig. 4B, KO mitochondria seem more swollen and clustered than elongated. It is

possible that because mitochondria are all clustered around large vacuoles (at least in this image), the clusters are picked up as one mitochondrion during segmentation.

We have revised this statement to reflect that the mitochondria are larger, the sum volume of mitochondria is increased, and there are more MDV-sized vesicles in both PFN1 KO CAD cells and MEFs. Mitochondria that are clustered are identified as a single object if the space between them cannot be resolved by super-resolution confocal microscopy. However, these larger clusters are a very small fraction of the often hundreds of objects measured per cell and do not influence median measurements of elongation and volume. Initially, we were also concerned that they could potentially skew the results, however excluding them from the analysis did not change any of the reported trends, so we decided not to filter the data.

15. What do the crossbars represent in the graphs? Average \pm SD? SEM? In several figures (obvious in Fig. 4D, F, I, N), the top and bottom bars are not equally distributed above and below what looks like the average. In any case, this needs to be clearly indicated in the figure legend.

All statistical parameters such as error bars, what each data point stands for, and what statistical test was used has been added to the figure legends.

16. Intuitively, one would think that the larger the mitochondrial network, the more fusion/fission there would be (assuming a constant number of fission/fusion events/ mitochondrial length). Any explanation as to why this is not the case in control cells? And what does this correlation actually say relative to mitochondrial dysfunction in KO cells? Or is this an issue with the quantification method?

Please see the response to concern #13.

17. Given that DRP1 only partially localizes to mitochondria where it forms foci, Pearson's correlation might not be the most sensitive measure. Given the very small difference observed between WT and KO cells, these results should be confirmed by calculating the actual number of DRP1 foci on mitochondria.

We have reanalyzed the data as suggested and now measure DRP1 as the number of foci on mitochondria, normalized against the area of the mitochondria that was measured so that cells with more mitochondria could be fairly compared to cells with less mitochondria. The same analysis method was used to measure MFN2 foci on mitochondria (Fig.4E-F).

Referee #3:

In this work, Read et al. use previously characterized PFN1 KO CAD cells to analyze the role of the actin-binding protein profilin-1 (PFN1) on mitochondrial ultrastructure and dynamics. The authors show that PFN1 loss leads to increased mitophagy and aberrant mitochondrial morphology associated with disturbed mitochondrial dynamics. Based on their findings, they propose the mitochondrial localization of PFN1 within the mitochondrial matrix and a new role for PFN1 in the overall regulation of mitochondrial metabolism and homeostasis. Overall, this work is mostly descriptive and lacks a fundamental molecular explanation connecting the observed findings. My main concern relates to the fact that the only parameter of mitochondrial function analyzed (respiration) is not clearly affected in PFN1 KO cells. I also have several concerns related to data interpretation, a number of points need clarification and some technical aspects of this work must be improved in order to reach the right conclusions.

Thank you for your thorough evaluation of our manuscript. We have addressed the technical and conceptual issues you raised in the individual responses below. In doing so, we have strengthened some of the conclusions made in the original draft of the manuscript and have modified others to better reflect the data which has come from new experiments and analysis.

- Comments to Figure 3: The authors claim unceasingly in the different sections of the manuscript that loss of PFN1 causes functional defects to mitochondria, or that it affects the functional integrity of mitochondria, but the only functional parameter measured relates to mitochondrial respiration (Figures 3A-F). And data interpretation concerning the effects of PFN1 loss on mitochondrial respiration is unconvincing, first because the reduction in mitochondrial basal respiration observed in PFN1 KO cells is very mild, and second because it does not even affect maximum respiration. It is therefore very unlikely that mitochondrial function and metabolism are heavily affected by PFN1 loss. Moreover, if there was a true defect in mitochondrial respiration one would expect to have higher ROS levels instead of decreased ROS, which is what happens in the PFN1 KO cells (Figure 3G). How can these inconsistencies be explained?

We have now included data showing that PFN1 KO mitochondria are depolarized (Fig. 3G) and have increased production of superoxide (Fig. 3H). Additionally, we have new data using mCherry-Parkin showing a mitochondria localization that is very similar to when mitochondria are depolarized with FCCP (compare Fig. 2B with Fig. 2C).

While the data using the CellRox probe was significant and reproducible, we have decided to remove it from the manuscript because we are not currently able to explain it. Our hypothesis is that the enzymatic systems which convert hydrogen peroxide, the ROS that CellRox predominately reports, to water have increased activity in PFN1 KO cells. While we didn't see increased expression of these enzymes in our RNAseq data, their activity could be increased at the protein level. Additionally, there is upregulation of several genes involved in peroxisome biogenesis and activity, which is consistent with an increased capacity to reduce cellular ROS (Fransen et al, BBA 2012). However, the CellRox result is not essential for the main points of the paper and the work required to determine the mechanisms of decreased cellular ROS is extensive. We look forward to investigating this more in a future study.

The results presented in Figure 3 in fact indicate that PFN1 KO cells display significantly decreased glycolytic rates than control cells, an adaptation occurring in the cytosol and not within mitochondria. And opposite to the authors' claims, glycolytic suppression normally reprograms intracellular energy metabolism towards mitochondrial OXPHOS (increased respiration) in multiple cell types. Maybe it would be an idea to revisit these data by incorporating the results previously obtained from RNAseq experiments regarding the relative expression of the enzymes related to energy metabolism (glycolysis, Krebs, etc), in order to gain some clues about the metabolic adaptations taking place in the KO cells and compare how these results match their functional observations.

An examination of the RNAseq data did not reveal anything telling about the change in expression of the genes involved in glycolysis. Although, a shift in metabolism could be occurring through changes occurring in this system at the protein level. Understanding the metabolic changes occurring in PFN1-deficient cells is a fantastic idea and something we would love to investigate in future studies. However, we would not be able to address it within the timeframe of this manuscript's resubmission.

- Figure 4C-L: Regarding ultrastructural parameters of mitochondrial morphology, the increase number of elongated mitochondria, number of objects and sum value in PFN1 KO cells could be attributed to an increased mitochondrial mass per cell, as a compensatory adaptation to the decreased glycolytic rates upon PFN1 loss. The authors need to normalize these parameters by mitochondrial mass.

When performing the analysis, we were sensitive to the potential issue that an increased number of objects could be caused by an increased amount of mitochondrial content. Regarding the two measurements where objects were counted in each cell, we did not find any influence of increased mitochondria content on the results. For example, in measurements of mitochondria $> 1 \mu\text{m}^3$, there was no difference in the results when we counted the number of objects this size per cell (Fig. 4C) to when we calculated it as a percentage of the total mitochondria volume (see below). For measurements of MDV-sized objects, we found no significant correlation between total mitochondria volume and the number of MDVs per cell that could bias the results (see below). Therefore, we have decided to keep the data as they are presented. Regarding the other two measurements, elongation describes the shape of an individual mitochondria and its measurement is not affected by total mitochondria content. Sum volume reports the total mitochondria content in each cell, assuming that mitochondria membrane localized TOM20 scales with increased amount of mitochondria. We rewritten the manuscript text and figure legends to prevent any further confusion about the measurements being made in Fig. 4 and Fig. EV3.

A curiosity, why would the CAD and MEF PFN1 KO cells behave different in terms of mitochondrial morphology?

We cannot say at the time, but uncovering why PFN1 loss of function causes mitochondria to elongate in CAD cells and enlarge in a way that makes them rounder in MEFs is something that we are excited to pursue in a future study. However, the fact that both CAD cells and MEFs have increased size of individual mitochondria (as well as sum mitochondria volume and number of MDV-sized objects, compare Fig. 4C and Fig. EV3F.), underscores the important role for PFN1 in maintaining mitochondria morphology.

- Figure 5: Why is there no increased mitochondrial fusion if there is less fission and decreased Drp1 levels? Please analyze other proteins involved in mitochondrial dynamics to have a clear picture of the alterations on mitochondrial dynamics, including Mfn1, Mfn2 or Opa1.

Due to a concern about the data coming from MitoMeter, we have removed the dynamics measurements obtained using that software from the manuscript. For more details about this concern, please see our response to Reviewer #2's concern #13. Instead, as you suggested, we performed the more conventional colocalization assays using DRP1 and MFN2. These results show reduced DRP1 and increased MFN2 on PFN1 KO mitochondria, indicating less fission and more fusion, which is consistent with the increased size and elongation of mitochondria in these cells (Fig. 4E-F).

- Figure 6 and manuscript text: Based on the presented data, it remains the possibility that PFN1 localizes attached to the IMM in the mitochondrial intermembrane space (IMS) rather than in the matrix. Several controls are required to convincingly prove the existence of PFN1 in the matrix (WB in Figure 6C): A positive matrix control (mtDNA, any Krebs cycle enzyme, etc), a positive IMS control (cytochrome c) and a negative control for cytosolic contamination. In addition the manuscript text is confusing, as sometimes the authors refer to the localization of PFN1 in the IMM and sometimes in the matrix.

Thank you for this suggestion. To address the localization of PFN1 to the mitochondria matrix, we performed the experiment of making mitoplasts and performing new assays with Proteinase K using the controls which you suggested on the western blot. As shown in Fig. 5C, PFN1 is protected from Proteinase K degradation even after the outer membrane was extracted with digitonin, as was the matrix protein citrate synthase. We now explicitly state that Profilin 1 is in the mitochondria matrix in the paper.

- General comment: Please add the specific statistical analysis used on each experiment in the figure legends.

All statistical parameters such as error bars, what each data point stands for, and what statistical test was used has been added to the figure legends.

Dear Eric,

Thank you for submitting your revised manuscript, which has been seen by all of the original referees. As you know, referees, especially #1 and #3, have outstanding concerns and I had already shared the reports with you. Thank you for sending me your preliminary response to these concerns. I have read them carefully and also got input from the referees on your response.

In brief, referees do not find that your proposed revision will satisfactorily address their concerns regarding the matrix localization of PFN1. Especially referee #3 states that "I agree that PFN1 is located in the mitochondria, but still not fully convinced of its matrix location. It could well be attached to the IMM as it behaves the same as COX4 in the left panel, and the authors omitted this antibody in the right panel so there is no way to differentiate between these possibilities unless all controls are used in both panels. This should be very simple to amend, by just probing those membranes with the antibodies against citrate synthase, COXIV and cytc in both panels." While referees #1 and #2 view these data more positively, referee #1 also finds that including all markers in these experiments would make it much more convincing. As such, given that matrix localization is a central claim of the study (as it is also present in the title), I would like to ask you either to include the markers referee #3 requests, or to de-emphasize the matrix localization, to rephrase the text as mitochondrial localization instead and to add a discussion point on the possible localization of PFN1 to the matrix or to the IMM.

Moreover, I need you to address the editorial points below before I can accept the manuscript.

- We believe that your study is better suited for our 'Scientific Report' format.
- Please provide 3-5 keywords for your study. These will be visible in the html version of the paper and on PubMed and will help increase the discoverability of your work.
- As per our guidelines, Data Availability section is reserved for the new primary dataset that is generated in this study and deposited in a public data repository. We note that dataset GSE135251 was not generated in this study. Therefore, please remove the reference to GSE149870. Instead, please cite GSE149870 in the text in the form of data citation - e.g. when Figures 1A and EV2A are called out as follows. Also, please cite the study separately as well.

In text: (Skruber et al, 2020; Data ref: Skruber et al, 2020).

In the reference list:

Dataset: Skruber K, Warp PV, Shklyarov R, Thomas JD, Swanson MS, Henty-Ridilla JL, Read TA, Vitriol EA (2020) Gene Expression Omnibus GSE149870 (<https://www.ncbi.nlm.nih.gov/geo/query/acc.cgi>). [DATASET]

Paper: Skruber K, Warp PV, Shklyarov R, Thomas JD, Swanson MS, Henty-Ridilla JL, Read TA, Vitriol EA (2020) Arp2/3 and Mena/VASP Require Profilin 1 for Actin Network Assembly at the Leading Edge. *Curr Biol* 20;30(14):2651-2664.e5.

Please see <https://www.embopress.org/page/journal/14693178/authorguide#referencesformat>

- Please rename the Declaration of Interests section as "Disclosure Statement and Competing Interests".
- We noted some name discrepancies between the manuscript text and the manuscript submission system (eJP) - i.e. Mitch T. Butler in the ms file vs, Mitchell T. Butler in eJP; Neil L. Weintraub in the ms file vs. Neal Weintraub in eJP
- Please fill out and include an author checklist as listed in our online guidelines (<https://www.embopress.org/page/journal/14693178/authorguide>)
- We note that ORCID iD of Dr. Tracy-Ann Read is currently not linked. EMBO Press policy asks for all corresponding authors to link to their ORCID iDs. You can read about the change under "Authorship Guidelines" in the Guide to Authors here: <https://www.embopress.org/page/journal/14693178/authorguide#authorshipguidelines>

In order to link your ORCID iD to your account in our manuscript tracking system, please do the following:

1. Click the 'Modify Profile' link at the bottom of your homepage in our system.
2. On the next page you will see a box halfway down the page titled ORCID*. Below this box is red text reading 'To Register/Link to ORCID, click here'. Please follow that link: you will be taken to ORCID where you can log in to your account (or create an account if you don't have one)
3. You will then be asked to authorise Wiley to access your ORCID information. Once you have approved the linking, you will be brought back to our manuscript system.

We regret that we cannot do this linking on your behalf for security reasons.

- We note that Figure 4G is currently not called out in the text.
- Please move the Figure Legends after the References.
- Materials and Methods should be renamed as Methods.
- The label A should be removed from Figures EV2 and EV4 as they contain one panel only. The text callouts should be updated

accordingly.

- Please resubmit the source data of the EV Figures only as a single zip file.
- Our production/data editors have asked you to clarify several points in the figure legends:
 - o Please note that a separate 'Data Information' section is required in the legends of figures 1b, d; 2b, d; 3b-c, e, g-i; 4c, e-g; EV 1a-b; EV 4b-c, f. Please see <https://www.embopress.org/page/journal/14693178/authorguide#figureformat> for an example.
 - o Please note that the figure title is missing for the figures EV 1, EV 2;EV4, EV5. This need to be rectified.
 - o Please note that there is no figure labelled as EV 3 in the manuscript. We are not sure if the figure is missing, or the labelling of the figures is incorrect. Kindly look into this.
 - o Please note that information related to n is missing in the legends of figures 3a, d.
 - o Please note that the error bars are not defined in the legend of figure EV 4a.
 - o Please note that the white arrows are not defined in the legend of figure 4e. This needs to be rectified.
- Papers published in EMBO Reports include a 'synopsis' and 'bullet points' to further enhance discoverability. Both are displayed on the html version of the paper and are freely accessible to all readers. The synopsis includes a short standfirst summarizing the study in 1 or 2 sentences (max 35 words) that summarize the paper and are provided by the authors and streamlined by the handling editor. I would therefore ask you to include your synopsis blurb and 3-5 bullet points listing the key experimental findings.
- In addition, please provide an image for the synopsis. This image should provide a rapid overview of the question addressed in the study but still needs to be kept fairly modest since the image size cannot exceed 550 (width) x 300-600 (height) pixels.

Thank you again for giving us to consider your manuscript for EMBO Reports, I look forward to your minor revision.

Kind regards,

Deniz

--

Deniz Senyilmaz Tiebe, PhD
Editor
EMBO Reports

Referee #1:

The authors have addressed most of my comments, and the manuscript now appears more developed, with a significant portion of the conclusions supported by data.

However, the issue of mitochondrial localization of PFN1, a crucial aspect of this work, still requires more substantial evidence to support it.

1. First, the 4xmts-mScarlet construct (Fig. 5D) is based on cytochrome c mitochondrial import sequence and, as such, is predicted to localize to mitochondrial intermembrane space. Intermembrane space markers (including overexpressed and endogenous cytochrome c) show diffuse localization within mitochondria, similar to matrix markers. They are not good reporters of the mitochondrial membranes. Furthermore, it needs to be clarified that this construct localizes in the IMS but is not stuck in the mitochondrial matrix due to abnormal repeats of the import sequence. Thus, the image in Fig. 5D likely shows 4xmts-mScarlet-positive mitochondria enveloping around GFP-PFN1-positive structures. But not localization of GFP-PFN1 inside the mitochondria. The authors' interpretation that GFP-PFN1 is in the matrix needs further experimental validation. These, and the fact that the subcellular localization pattern of GFP-PFN1 resembles ER, warrants additional experiments.

2. The proteinase K experiments, while informative, lack specific markers of distinct submitochondrial compartments, including matrix and IMM. For instance, cytochrome c is an IMS (only partially loosely attached to the IMM) protein and is expected to be lost from mitochondrial fractions even in triton-treated samples without adding proteinase K. Whole cell lysate is also not shown in the right panel (Digitonin-treated samples). All proteins should be blotted in all samples shown (e.g., please show Tom20 in the right-side panels, and Citrate synthase in the left-side blots in Fig. 5C). These limitations highlight the need for additional validation and more specific markers.

3. Another point that needs additional experimental evaluation is the localization of mCherry-Parkin to the mitochondria. The Parkin localization pattern is intriguing, but is Parkin associated with mitochondria in the images shown in Figures 2B and 2C? The pattern is distinct enough from mitochondria, so the authors should use additional mitochondrial markers (e.g., mitochondrial matrix protein) to verify Parkin and p62 localization in DMSO-treated PFN1 ko cells.

Referee #2:

The revised version of the manuscript by Read et al. properly addresses my concerns. I only have a few small issues remaining:

1. The labeling of Fig. 2D indicates Lat A. Were these cells treated with Lat A? This would be an important issue as the experiment is supposed to address mitophagy in control vs KO cells, not the effect of actin depolymerization.
2. Line 180, there is a word missing in this sentence.

3. The MFN2 images that are shown seem heavily processed. It is thus unclear whether this really represents the actual staining. While MFN2 does form foci, some level of mitochondrial staining should be present.
4. Line 315, the part on mitochondrial dynamics (and reference to the figures - Fig 4G is not mentioned) should be updated to better reflect that mitochondrial dynamics have not been directly measured.

Referee #3:

My concerns have been mostly addressed by the authors, except for one relevant issue that needs to be convincingly proved in order to support the author's conclusions (even affecting the title of the manuscript). The matrix location of PFN1 remains very unconvincing and data interpretation from the new figure 5C is just wrong. I agree from the PK treatment results that PFN1 is internalized within mitochondria and not located in the OMM, but the data provided do not exclude either its IMS or IMM location, rather than being located in the matrix. Moreover, the different concentrations of digitonin used in figure 5C before PK treatment are not sufficient to completely solubilize the IMM, as shown in multiple BNE analyses found in the literature showing the presence of respiratory chain complexes in purified mitochondria treated with those solubilisation conditions. These concentrations of digitonin are not even sufficient to completely disrupt the OMM, as VDAC or TOM complexes can also be easily addressed by BNE of mitochondria solubilised with 4g/g detergent/protein. Mitoplasts can be considered as "permeabilised" mitochondria, but still contain a substantial amount of OMM and IMS attached to the IMM. All these experiments, either using triton X-100 or digitonin, thus need to include at least one protein for each one of the mitochondrial subfractions (OMM, IMM, IMS and matrix) for an accurate determination of the PFN1 location. By the way, cytc is an IMS protein, not an IMM (as stated in page 8 of the manuscript text).

Dear Dr. Senyilmaz,

Here is our point-by-point response to the Referee comments, with our responses in blue. With a quick revision and the inclusion of some additional data, we can address all the concerns which have been raised by Referees #1 and #3, which largely stem from an incomplete understanding of the experiments we performed. In the final revision, we will make sure that the text and figures are as clear as possible to prevent any further confusion. There is some data that we have included in this response that, in retrospect, should have been included with the resubmitted version of the manuscript to preemptively address the concerns which were raised. The final submission will include this data. Thank you for your time and the opportunity to send a rebuttal.

Best,
Eric Vitriol

Referee #1:

The authors have addressed most of my comments, and the manuscript now appears more developed, with a significant portion of the conclusions supported by data.

However, the issue of mitochondrial localization of PFN1, a crucial aspect of this work, still requires more substantial evidence to support it.

1. First, the 4xmts-mScarlet construct (Fig. 5D) is based on cytochrome c mitochondrial import sequence and, as such, is predicted to localize to mitochondrial intermembrane space. Intermembrane space markers (including overexpressed and endogenous cytochrome c) show diffuse localization within mitochondria, similar to matrix markers. They are not good reporters of the mitochondrial membranes. Furthermore, it needs to be clarified that this construct localizes in the IMS but is not stuck in the mitochondrial matrix due to abnormal repeats of the import sequence. Thus, the image in Fig. 5D likely shows 4xmts-mScarlet-positive mitochondria enveloping around GFP-PFN1-positive structures. But not localization of GFP-PFN1 inside the mitochondria. The authors' interpretation that GFP-PFN1 is in the matrix needs further experimental validation. These, and the fact that the subcellular localization pattern of GFP-PFN1 resembles ER, warrants additional experiments.

First, we would agree with this referee in that this probe should have been labeled as an intermembrane space (IMS) marker. That being said, the IMS is so small (10-20 nm; see Mageswaran et al., 2023) that at the level of light microscopy, including in 4X expanded samples, an IMS probe is indistinguishable from an IMM probe. The fluorescence images clearly show GFP-PFN1 associated with TOM20 labeled mitochondria (Fig. 5A), ALS-PFN1 aggregates inside of the mitochondria OMM (Fig. 5B), and GFP-PFN1 is inside the IMS (Fig. 5D, note the closed 4xmts-mScarlet compartments in the enlarged insets, which correspond to negative fluorescence in the GFP-PFN1 channel. These images were all taken from single planes of confocal z-stack). By themselves, we would agree that this is not enough to determine the location of PFN1 in mitochondria. In fact, all of this data could be removed from the figure, or placed into the associated EV figure, without altering Fig. 5's main message, where we explicitly show endogenous PFN1 in the matrix, which is detailed below.

2. The proteinase K experiments, while informative, lack specific markers of distinct submitochondrial compartments, including matrix and IMM.

The article does include specific markers of different cells and submitochondrial compartments, selected according to literature information and reviewer requests:

- GAPDH is a marker for cytosol (**Referee #3 requested**).
- TOM20 is a specific marker for OMM.
- COXIV is a specific marker for IMM.
- Cytochrome *c* is a specific marker for IMS (**Referees #2 and #3 requested**).
- Citrate synthase is a specific marker for the mitochondrial matrix (**Referee #3 requested**).

For instance, cytochrome *c* is an IMS (only partially loosely attached to the IMM) protein and is expected to be lost from mitochondrial fractions even in triton-treated samples without adding proteinase K.

Using traditional methods of lysing cells to purify mitochondria, the integrity of the OMM in the isolated mitochondria will indeed be compromised, in which case the cytochrome *c* “*is expected to be lost from mitochondrial fractions even in triton-treated samples without adding proteinase K*” as the referee #1 says. However, we dedicated substantial time and resources to finding and standardizing the appropriate protocol for obtaining a significant number of mostly intact mitochondria, a mandatory condition for accurately studying the localization of a protein at a specific submitochondrial compartment. The low-pressure nitrogen cavitation used in this article disrupts the plasma membrane without subjecting mitochondria to shear stress, leaving them fully intact and functional (**Gottlieb and Adachi, 2000**). This allows the isolation of a mitochondrial fraction positive for TOM20, cytochrome *c*, COX IV and citrate synthase, and negative for GAPDH.

Whole cell lysate is also not shown in the right panel (Digitonin-treated samples). All proteins should be blotted in all samples shown (e.g., please show Tom20 in the right-side panels, and Citrate synthase in the left-side blots in Fig. 5C). These limitations highlight the need for additional validation and more specific markers.

The Fig. 5C experiments are two independent experiments that answer complementary but different questions, requiring different and specific markers/controls.

On the left side, the PFN1 cellular localization was analyzed by comparing cytosolic (Cyto) vs. mitochondrial (Mito) fractions. For this, we used markers for cytosol (PFN1), OMM (TOM20), and IMM (COX IV). In short, the Cyto fraction (PFN1 positive) and Mito fraction (TOM20, COX IV, and PFN1 positive) were obtained. The Mito fraction was treated with Proteinase K resulting in TOM20 negative and COXIV and PFN1 positive. Indicating that Proteinase K eliminated proteins from the outer surface of OMM but did not have access to proteins located inside the mitochondria. To complement the above, a Mito fraction was first permeabilized with Triton X-100 and then treated with Proteinase K; in that case, the Proteinase K degraded COX IV and PFN1. The results allow us to conclude that PFN1 is present inside the OMM. Including citrate synthase or *cyt c* in this experiment would not add more information to the conclusion. For better clarity, we will change the experiment's name from "OMM vs. IMM PFN1 localization" to a more precise one, for example: “Outside vs. Inside the OMM.”

On the right side, the PFN1 intramitochondrial localization was analyzed by comparing IMS vs. matrix. For this, we used cytosol markers (GAPDH and PFN1), IMS markers (Cyt c), and matrix marker (Citrate synthase). In short, the Cyto fraction (GAPDH and PFN1 positive), and Mito fraction (Cyt c and Citrate synthase positive) were obtained. The Mito fraction was treated with digitonin to eliminate/permeabilize the OMM, obtaining mitoplasts (MP). That mitoplasts were obtained that still included IMM-associated cytochrome c is a testament to the gentle methods we used to isolate mitochondria and carefully remove the OMM. Western blotting for TOM20 was used during the mitoplast generation process (see below), and we will add this data to the EV of this figures. Mitoplasts were treated with Proteinase K, resulting in a notably decreased Cyt c level compared to the Mito fraction, while Citrate synthase and PFN1 levels remained similar. Indicating that Proteinase K degraded proteins from IMS but did not have access to proteins located inside the mitochondria. The results allow us to conclude that PFN1 is present in the mitochondrial matrix.

It is also important to note that we are using enough Proteinase K in these assays to digest all of the PFN1 in the cytoplasmic fraction (see below), and this is a protein that is expressed at concentrations of 50-100 μ M (Skruber et al, 2018).

For better clarity, we can add the following data to Figure EV5:

Data to be added to Figure EV5.

(A) Proteinase K (PK) titration. Western blot for PFN1 in Cytoplasmic fraction (Cyto), and Cyto treated with 0.5, 1, 3, or 5 mg/mL PK for 1 hour at 37°C.

(B) Mitoplast generation. Western blot for citrate synthase and TOM20 in mitochondrial fraction (Mito) treated with 0.5, 1, 2, or 4 mg digitonin/mg mitochondrial proteins for 20 mins at 4°C at 2,000 rpm, and Mito.

To conclude, I will quote Referee #1's previous comments from the review of the first submission. *“The authors could expand these studies and use osmotic swelling to open the OMM and apply the proteinase K treatment to the swollen mitochondria. Under these conditions, PFN1 should not be significantly degraded only if it localizes in the matrix but would be degraded if present in the intermembrane space or on the outer surface of the IMM”.*

Following their requests, our results can be described as follows: We expanded our studies and used digitonin to solubilize the OMM and applied proteinase K to the mitoplasts. Under these conditions, PFN1 was not significantly degraded because it was localized in the matrix and was not present in the intermembrane space or on the outer surface of the IMM.

Finally, it should be noted that another group has used the data in our preprint to validate the findings of their new approach for localized proteomics in mitochondria subcompartments and have shown that PFN1 is inside mitochondria in HEK 293 cells (Zhu et al., 2024). This was published during the second review of our manuscript. Using their method, they see both IMS and matrix localized PFN1, though they used more traditional purification techniques and osmotic swelling to burst the outer membrane, which are arguably not as clean as the methods we used in this paper. Our data shows absolutely no loss of PFN1 upon Proteinase K treating mitoplasts, but almost complete loss of cytochrome C, revealing that all of the PFN1 in our mitochondria fractions is protected from Proteinase K because they are inside the IMM. We will update the final submission to include this reference and discuss it. That another group has independently verified our findings in cells from a different species and tissue, citing our preprint as the inspiration to look at PFN1, strengthens the conclusions that are made in our paper.

3. Another point that needs additional experimental evaluation is the localization of mCherry-Parkin to the mitochondria. The Parkin localization pattern is intriguing, but is Parkin associated with mitochondria in the images shown in Figures 2B and 2C? The pattern is distinct enough from mitochondria, so the authors should use additional mitochondrial markers (e.g., mitochondrial matrix protein) to verify Parkin and p62 localization in DMSO-treated PFN1 ko cells.

In general, mCherry-Parkin is localized to mitochondria. There are some small areas where the Parkin foci do not completely overlap with TOM20, but this could be interpreted as Parkin-labeled mitochondria that are in acidified lysosomes, where the TOM20 epitope for the antibody has been degraded but the mCherry signal is still functional. The main point of that figure is that mitophagy has been activated, which the sum of the data clearly shows.

Referee #2:

The revised version of the manuscript by Read et al. properly addresses my concerns. I only have a few small issues remaining:

1. The labeling of Fig. 2D indicates Lat A. Were these cells treated with Lat A? This would be an important issue as the experiment is supposed to address mitophagy in control vs KO cells, not the effect of actin depolymerization.

This was an error, there was no Lat A used for those experiments. It will be corrected in the final submission.

2. Line 180, there is a word missing in this sentence.

This will be corrected in the final submission.

3. The MFN2 images that are shown seem heavily processed. It is thus unclear whether this really represents the actual staining. While MFN2 does form foci, some level of mitochondrial staining should be present.

The images are processed identically to other super-resolution images seen throughout the paper, only using denoise.Ai to remove camera noise and deconvolution to enhance the resolution. The images have been scaled to emphasize the puncta which were quantified in the graph. We can change the LUT values slightly in the final submission to make the image look “less processed”.

4. Line 315, the part on mitochondrial dynamics (and reference to the figures - Fig 4G is not mentioned) should be updated to better reflect that mitochondrial dynamics have not been directly measured.

This will be corrected in the final submission.

Referee #3:

My concerns have been mostly addressed by the authors, except for one relevant issue that needs to be convincingly proved in order to support the author’s conclusions (even affecting the title of the manuscript). The matrix location of PFN1 remains very unconvincing and data interpretation from the new figure 5C is just wrong. I agree from the PK treatment results that PFN1 is internalized within mitochondria and not located in the OMM, but the data provided do not exclude either its IMS or IMM location, rather than being located in the matrix.

This concern is addressed in our response to Referee #1.

Moreover, the different concentrations of digitonin used in figure 5C before PK treatment are not sufficient to completely solubilize the IMM, as shown in multiple BNE analyses found in the literature showing the presence of respiratory chain complexes in purified mitochondria treated with those solubilisation conditions.

Digitonin was used to permeabilize the OMM and allow Proteinase K access to IMS, not to completely solubilize the IMM. The article repeatedly describes the reason for using digitonin as a permeabilizing agent for the OMM to generate mitoplasts. Likewise, the use of digitonin for such use is described in the literature (*Sileikyte et al., 2011; van Vlies et al., 2007*).

These concentrations of digitonin are not even sufficient to completely disrupt the OMM, as VDAC or TOM complexes can also be easily addressed by BNE of mitochondria solubilised with 4g/g detergent/protein. Mitoplasts can be considered as "permeabilised" mitochondria, but still contain a substantial amount of OMM and IMS attached to the IMM.

We did not aim to eliminate the OMM completely with digitonin; as seen above in our response to Referee #1, the levels of TOM20 do not completely disappear. We only permeabilized the OMM enough to allow the entry of Proteinase K to the IMS. This approach notably degraded cyt c in mitoplast treated with Proteinase K while leaving citrate synthase and PFN1 unaffected. It should be noted that we were able to generate mitoplasts that still contained cyt c, as this complex is only weakly attached to the IMM. This speaks to the care in which mitochondria were purified and the outer membrane was permeabilized.

Throughout the course of our experiments, we evaluated different digitonin concentrations and conditions described in the literature. For instance, Sileikyte and colleagues (2011) obtained mitoplasts (90% decrease in monoamine oxidase activity) from rat livers using the following protocol: 0.09–0.12 mg of digitonin/mg of mitochondrial protein in an ice-water bath and gently stirred for 20 min. We did not observe differences in TOM20 levels in western blot following these concentrations and conditions (**Appendix 1A**). On the other hand, van Vlies and colleagues (2007) obtained mitoplasts (65% release of adenylate kinase and only 3% release of citrate synthase) from rat kidneys using the following protocol: 1.25 mg digitonin/mg de mitochondrial protein at 4°C for 10 min. We did not observe differences in TOM20 levels in western blot either following these concentrations and conditions (**Appendix 1B**). After many attempts, we observed a significant decrease in TOM20 level using 4 mg digitonin/mg of mitochondrial protein under intense agitation (2,000 rpm) at 4°C for 20 min (**Appendix 1C**).

Appendix 1. Different protocols evaluated for the Mitoplast generation.

All these experiments, either using triton X-100 or digitonin, thus need to include at least one protein for each one of the mitochondrial subfractions (OMM, IMM, IMS and matrix) for an accurate determination of the PFN1 location. By the way, cytc is an IMS protein, not an IMM (as stated in page 8 of the manuscript text)

In the words of Referee #3 from the first review. *“Several controls are required to convincingly prove the existence of PFN1 in the matrix (WB in Figure 6C): A positive matrix control (mtDNA, any Krebs cycle enzyme, etc), a positive IMS control (cytochrome c) and a negative control for cytosolic contamination”*

Following the guidance of this and other referees, the article includes specific markers for different cells and submitochondrial compartments:

- GAPDH is a marker for cytosol (**Referee #3 requested**).
- TOM20 is a specific marker for OMM.
- COXIV is a specific marker for IMM.
- Cytochrome c is a specific marker for IMS (**Referees #2 and #3 requested**).
- Citrate synthetase is a specific marker for the mitochondrial matrix (**Referee #3 requested**).

References

Gottlieb RA, Adachi S (2000) Nitrogen cavitation for cell disruption to obtain mitochondria from cultured cells. *Methods Enzymol* 322: 213-221.

Mageswaran SK, Grotjahn DA, Zeng X, Barad BA, Medina M, Hoang MH, Dobro MJ, Chang YW, Xu M, Yang WY, Jensen GJ. Nanoscale details of mitochondrial constriction revealed by cryoelectron tomography. *Biophys J*. 2023 Sep 19;122(18):3768-3782.

Sileikyte J, Petronilli V, Zulian A, Dabbeni-Sala F, Tognon G, Nikolov P, Bernardi P, Ricchelli F (2011) Regulation of the inner membrane mitochondrial permeability transition by the outer membrane translocator protein (peripheral benzodiazepine receptor). *J Biol Chem* 286: 1046-1053.

Skruber K, Read TA, Vitriol EA. Reconsidering an active role for G-actin in cytoskeletal regulation. *J Cell Sci*. 2018 Jan 10;131(1):jcs203760.

van Vlies N, Ofman R, Wanders RJ, Vaz FM (2007) Submitochondrial localization of 6-N-trimethyllysine dioxygenase - implications for carnitine biosynthesis. *FEBS J* 274: 5845-5851.

Zhu Y, Akkaya KC, Ruta J, Yokoyama N, Wang C, Ruwolt M, Lima DB, Lehmann M, Liu F. Cross-link assisted spatial proteomics to map sub-organelle proteomes and membrane protein topologies. *Nat Commun*. 2024 Apr 17;15(1):3290.

Dr. Eric Vitriol
Medical College of Georgia at Augusta University
Neuroscience & Regenerative Medicine
1120 15th St.
Augusta, GA 30912
United States

Dear Eric,

Thank you for submitting your revised manuscript. I have now looked at everything and all is fine. Therefore, I am very pleased to accept your manuscript for publication in EMBO Reports.

Congratulations on a nice work!

Kind regards,

Deniz
--
Deniz Senyilmaz Tiebe, PhD
Editor
EMBO Reports
